# CXCR5 CAR-T cells simultaneously target B cell non-Hodgkin's lymphoma and tumor-supportive follicular T helper cells

Mario Bunse [1,7], Janina Pfeilschifter[1,7], Julia Bluhm[2], Maria Zschummel[1], Jara J. Joedicke [1], Anthea Wirges[2], Helen Stark [2], Vivien Kretschmer[1], Markus Chmielewski[3], Wolfgang Uckert[4], Hinrich Abken [5], Jörg Westermann[6], Armin Rehm [2✉] & Uta E. Höpken [1✉]

CAR-T cell therapy targeting CD19 demonstrated strong activity against advanced B cell leukemia, however shows less efficacy against lymphoma with nodal dissemination. To target both B cell Non-Hodgkin's lymphoma (B-NHLs) and follicular T helper (Tfh) cells in the tumor microenvironment (TME), we apply here a chimeric antigen receptor (CAR) that recognizes human CXCR5 with high avidity. CXCR5, physiologically expressed on mature B and Tfh cells, is also highly expressed on nodal B-NHLs. Anti-CXCR5 CAR-T cells eradicate B-NHL cells and lymphoma-supportive Tfh cells more potently than CD19 CAR-T cells in vitro, and they efficiently inhibit lymphoma growth in a murine xenograft model. Administration of anti-murine CXCR5 CAR-T cells in syngeneic mice specifically depletes endogenous and malignant B and Tfh cells without unexpected on-target/off-tumor effects. Collectively, anti-CXCR5 CAR-T cells provide a promising treatment strategy for nodal B-NHLs through the simultaneous elimination of lymphoma B cells and Tfh cells of the tumor-supporting TME.

[1] Max-Delbrück-Center for Molecular Medicine, MDC, Department of Microenvironmental Regulation in Autoimmunity and Cancer, 13125 Berlin, Germany. [2] Max-Delbrück-Center for Molecular Medicine, MDC, Department of Translational Tumorimmunology, 13125 Berlin, Germany. [3] Department I of Internal Medicine, University Hospital Cologne, 50931 Cologne, Germany. [4] Max-Delbrück-Center for Molecular Medicine, MDC, Department of Molecular Cell Biology and Gene Therapy, 13125 Berlin, Germany. [5] RCI Regensburg Center for Interventional Immunology, Department Genetic Immunotherapy, University Hospital Regensburg, 93053 Regensburg, Germany. [6] Department of Hematology, Oncology and Tumorimmunology, Campus Virchow Klinikum; Charité-University Medicine, 13353 Berlin, Germany. [7]These authors contributed equally: Mario Bunse, Janina Pfeilschifter. ✉email: arehm@mdc-berlin.de; uhoepken@mdc-berlin.de

Bcell non-Hodgkin's lymphoma (B-NHL) is a heterogenous group of clonal neoplasia that preferentially home to secondary lymphoid organs (SLOs)[1,2]. Despite improved progression-free and overall survival for patients with aggressive B-NHL by standard chemotherapy and the addition of rituximab, about 30–40% of patients relapse early or become refractory[3]. After the failure of front-line treatments the prognosis decreases dramatically[4]. In patients with diffuse large B cell lymphoma (DLBCL) who progressed after high-dose chemotherapy with autologous stem-cell transplantation, the median overall survival is <10 months[5,6]. The rapid emergence of secondary resistances to targeted therapies in particular in nodal diseases urges to find novel therapeutic strategies.

Adoptive transfer of chimeric antigen receptor (CAR)-T cells targeting antigens on lymphoid cells has revolutionized the treatment of advanced B cell malignancies[7,8]. Foremost anti-CD19 CAR-T cell therapies have proven remarkable efficacy in B-cell acute lymphoblastic leukemia (B-ALL), chronic lymphocytic leukemia (CLL)[9–11], and B-NHL[12–15], however, these CARs can become ineffective due to CD19 antigen loss or downregulation[16,17]. Additional difficulties in the treatment of nodal lymphoma in comparison to the elimination of circulating leukemic cells arise from the specific suppressive environment that reduces CAR-T cell anti-lymphoma efficacy[18,19].

To improve the treatment efficacy of nodal B-NHLs we aimed at targeting both the lymphoma cells and the tumor-supportive follicular T-helper (Tfh) cells within the nodal stroma. We identified the chemokine receptor CXCR5 as a superior target for CAR-T cell therapy since CXCR5 is expressed on mature B cells, their malignant counterparts, and on Tfh cells[2,20,21], but not on B cell precursors within the BM, or on plasma cells. Physiologically, CXCR5 regulates homeostatic lymphoid cell trafficking and homing to B cell follicles within SLOs[20]. Together with members of the lymphotoxin/tumor necrosis factor family, CXCR5 and its ligand CXCL13 are crucial for the development and maintenance of SLOs. Accordingly, CXCR5-deficient mice show impaired B cell follicle development[20,22]. B-NHL with nodal lodgings, such as follicular lymphoma (FL), DLBCL, mantle cell lymphoma (MCL), and CLL recapitulate the conserved lymphoid dissemination pattern associated with CXCR5 expression[2,23–25]. In a xenograft lymphoma mouse model, continuous anti-CXCR5 antibody treatment retarded tumor growth[26]. In a murine CLL model, CXCR5-deficiency impaired nodal homing and profoundly reduced lymphoma progression. The tumor-supportive CXCR5 interaction prompted us to interrogate CXCR5 as a target aiming at disrupting the CXCR5-dependent lymphoma cell dissemination and tumor-stroma interaction[27].

Here, we report on anti-human CXCR5 CAR-T cells that mediate killing of B-NHLs and concomitantly, eradicate CXCR5+ Tfh cells. The anti-CXCR5 CAR endows T cells with high avidity, necessary for anti-tumor efficacy in vitro and in vivo without conferring unintended reactivity against various non-hematopoetic cell types. Anti-murine CXCR5 CAR-T cells exhibit potent anti-tumor activity in vitro, and most importantly, facilitate specific benign and malignant B and CXCR5+ Tfh cell depletion without further off-target activity in a syngeneic mouse model. Our data suggest CXCR5 targeting for improved immunotherapy of nodal B-NHL by eliminating lymphoma-associated Tfh cells along with the lymphoma cells.

## Results

### Exploiting the chemokine receptor CXCR5 as a B-NHL-selective target.
We explored targeting CXCR5 as a novel therapeutic option for patients with mature B-NHL. Strong CXCR5 expression was detected on B-NHL cell lines DOHH-2 and SC-1

(FL), SU-DHL4 and OCI-Ly7 (germinal center type (GC)B-DLBCL), and JeKo-1 (MCL) (Fig. 1a). The B-ALL and multiple myeloma (MM) cell lines (NALM-6, REH, NCI-H929) derived from precursor B and plasma cells, respectively, lacked CXCR5 expression as does the T-ALL cell line Jurkat. By Quantibrite quantification, the highest CXCR5 antigen density was found on DOHH-2 cells (5146 receptors per cell), followed by SU-DHL4 (2846), SC-1 (2602), JeKo-1 (1667), and OCI-Ly7 (418) cells (Fig. 1b). A representative number of patient-derived primary lymphoma cells from FL, CLL, MCL, and marginal zone lymphoma (MZL) were analyzed for the level of CXCR5 surface expression. FL B cells exhibited robust CXCR5 expression (1602-2604 molecules per cell), except for one out of seven samples with low CXCR5 levels (#85: 265 molecules). Eight B-CLL patient samples uniformly showed the highest CXCR5 expression with 3063 up to 7159 molecules per cell, whereas CXCR5 expression in six MCL patient samples varied from low to high levels (329-4633 molecules). MZL B cells expressed minor numbers of CXCR5 molecules, only in one case a substantial CXCR5 expression was detected (69-100; #728: 2425 molecules per cell) (Fig. 1c, d and Supplementary Fig. 1a–d and Fig. 2a–d).

High expression levels of CXCR5 particularly in FL, CLL, and MCL cells were confirmed by a gene expression database analysis containing 630 cancer categories (HS_AFFY_U133PLUS_2.0; cancer entities with highest CXCR5 expression levels are shown in Supplementary Fig. 3)[28]. We confirmed that benign CD19+ B cells derived from peripheral blood leukocytes (PBL) expressed high amounts of CXCR5 molecules (Fig. 1e and Supplementary Figs. 1e and 2e). We also detected CXCR5 expression on a subpopulation of PBL-derived CD4+ T cells including circulating CD4+CXCR5+PD-1+ Tfh cells as well as on a minor subpopulation of CD8+ T cells (Supplementary Fig. 4a, c). CXCR5 antigen densities in CD4+ and CD8+ T cells were in the range of 1000-2000 molecules per cell (Fig. 1e). In contrast, CXCR5 was not detected on PBL-derived NK cells (CD3-CD56+), CD14+ monocytes, CD11c+HLA-DR+CD1c+ mDC1 cells, or on CD11c+HLA-DR+CD303+ pDC cells (Supplementary Fig. 4b, d–f). Collectively, the restricted surface expression on mature B-NHL entities and on Tfh cells suggests CXCR5 as a target for CAR-T cell therapy with the anticipated side effect of eliminating benign mature B cells.

### The anti-CXCR5 CAR retargets T cells specifically against CXCR5-expressing tumor cell lines.
We engineered the CXCR5 CAR based on the monoclonal rat anti-human CXCR5 antibody (mAb) RF8B2[29,30] that exhibits a binding affinity of KD = 0.7 nM. We localized the binding epitope of the RF8B2 mAb to aa 9-30 in the extracellular domain of CXCR5 using Jurkat cell lines expressing either the full receptor, or variants with deletions in the extracellular N-terminal domain (Supplementary Fig. 5a). A humanized second-generation anti-human CXCR5-28ζ CAR (referred to as CXCR5 CAR) was constructed similarly to our BCMA CAR[31] (Fig. 2a). After retroviral transduction the CAR was expressed by human T cells at a frequency of 54.9 ± 12.7% cells compared with 39.2 ± 12.79% cells expressing the SP6 (control) CAR (Fig. 2b, c). The CD8 to CD4 CAR-T cell ratios were similar for the CXCR5 CAR (64.4 ± 8.1% to 35.4 ± 9.4%) and for the SP6 CAR (61.5 ± 11.8% to 38.5 ± 11.9%) (Fig. 2d). The viral copy number (VCN) at mean values were <3 retroviral copies per transduced T cell (CXCR5 CAR: 2.03 ± 0.19; SP6 CAR: 2.24 ± 0.24), thus representing an acceptable risk profile (Fig. 2e). Stimulation of the CAR-T cells with clonal cell lines expressing graded amounts of CXCR5 demonstrated a sufficiently high avidity for CXCR5 levels found on primary B-NHLs and CXCR5-positive lymphocytes (Supplementary Fig. 5b) thereby making the

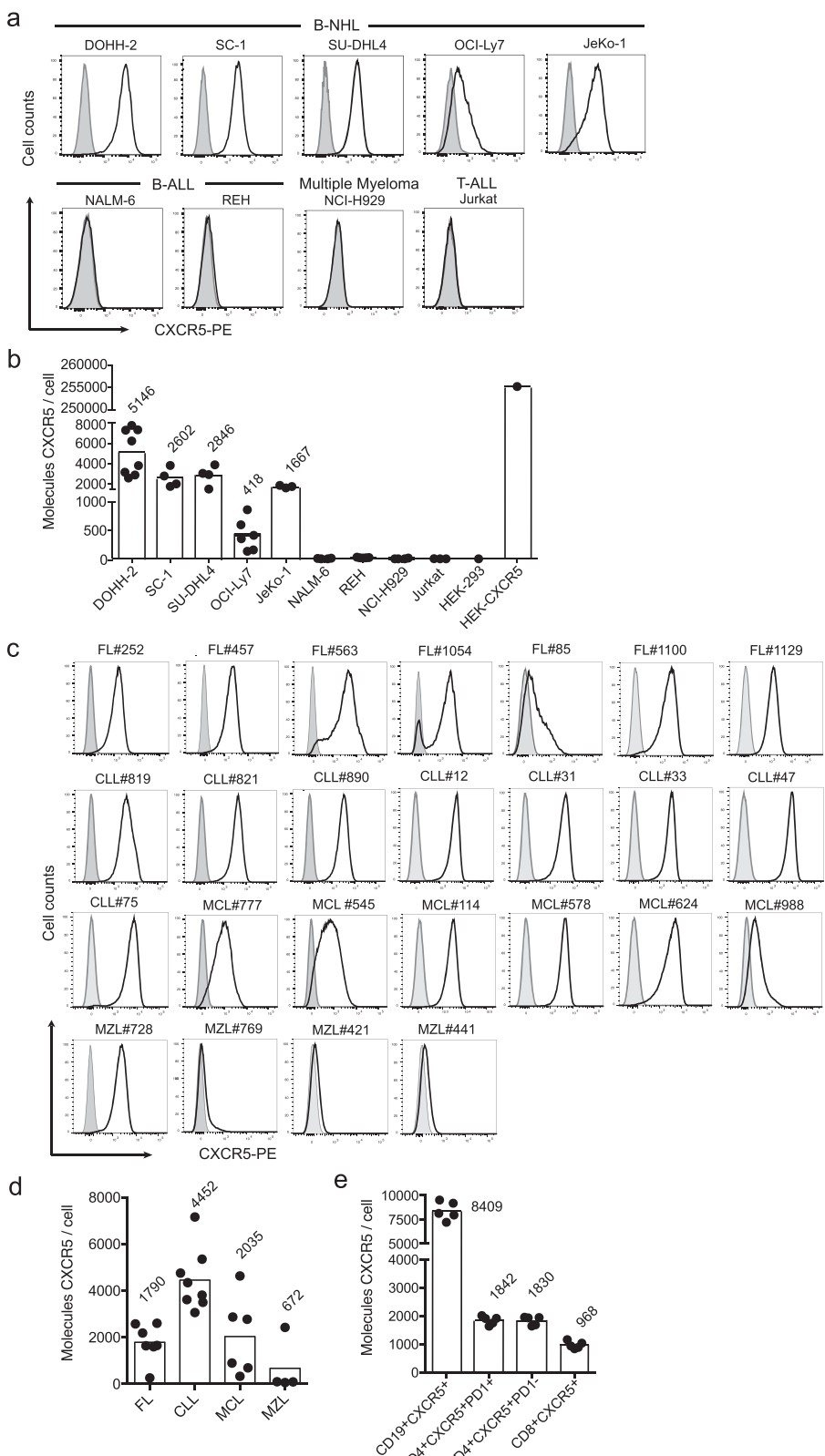

CXCR5 CAR suitable for lymphoma targeting. The EC50 for CAR induced IFNγ secretion was 900 molecules per cell; even <200 molecules triggered 18% of the maximal response.

Since CXCR5 is transiently upregulated on a T cell subpopulation during activation[30], we asked whether manufacturing of clinically relevant CAR-T cell numbers is feasible despite potential fratricide by CAR-T cells. Using a small-scale simulation model of the CliniMACS Prodigy (Miltenyi Biotec) bioreactor process we also recorded a transient upregulation of CXCR5 on activated T cells but no CXCR5 expression was detected at the end of the expansion phase (day 13). The total yield of genetically modified T cells over a 13-day cultivation period was 25–30% lower compared to samples transduced with the SP6 CAR (Supplementary Fig. 6). The successful manufacturing demonstrates that CXCR5 CAR-T cells

**Fig. 1 CXCR5 is strongly expressed on mature B-NHL cell lines and primary B-NHL cells.** Surface expression of CXCR5 was analyzed by flow cytometry and the number of CXCR5 molecules per cell was determined for tumor cell lines (**a**, **b**), patient-derived primary B-NHL cells (**c**, **d**) and for peripheral blood lymphocytes (**e**). Representative data (**a**; CXCR5, black line; isotype control, filled curve) and pooled data (**b**) from independent measurements of the following tumor cell lines: B-NHL (FL: DOHH-2 $n = 8$, SC-1 $n = 4$; DLBCL: SU-DHL4 $n = 4$, OCI-Ly7 $n = 6$; MCL: JeKo-1 $n = 3$), B-ALL (NALM-6 $n = 3$, REH $n = 4$), MM (NCI-H929 $n = 3$), T-ALL (Jurkat $n = 2$) and controls (HEK $n = 1$, HEK-CXCR5 $n = 1$). CXCR5 surface expression (**c**; CXCR5, black line; isotype control, filled curve) and density per cell (**d**) of 25 patient-derived primary B-NHL cells (FL $n = 7$: #252, #457, #563, #1054, #85, #1100, and #1129; CLL $n = 8$: #819, #821, #890, #12, #31, #33, #47 and #75; MCL $n = 6$: #777, #545, #114, #578, #624, and #988; MZL $n = 4$: #728, #769, #421, and #441). The gating strategies are shown in Supplementary Figs. 1a–d, 2a–d. **e** CXCR5 density per cell for peripheral blood lymphocyte subpopulations of $n = 5$ healthy individual donors. **b**, **d**, **e** Individual values and means as bars. Source data are provided as a Source Data file.

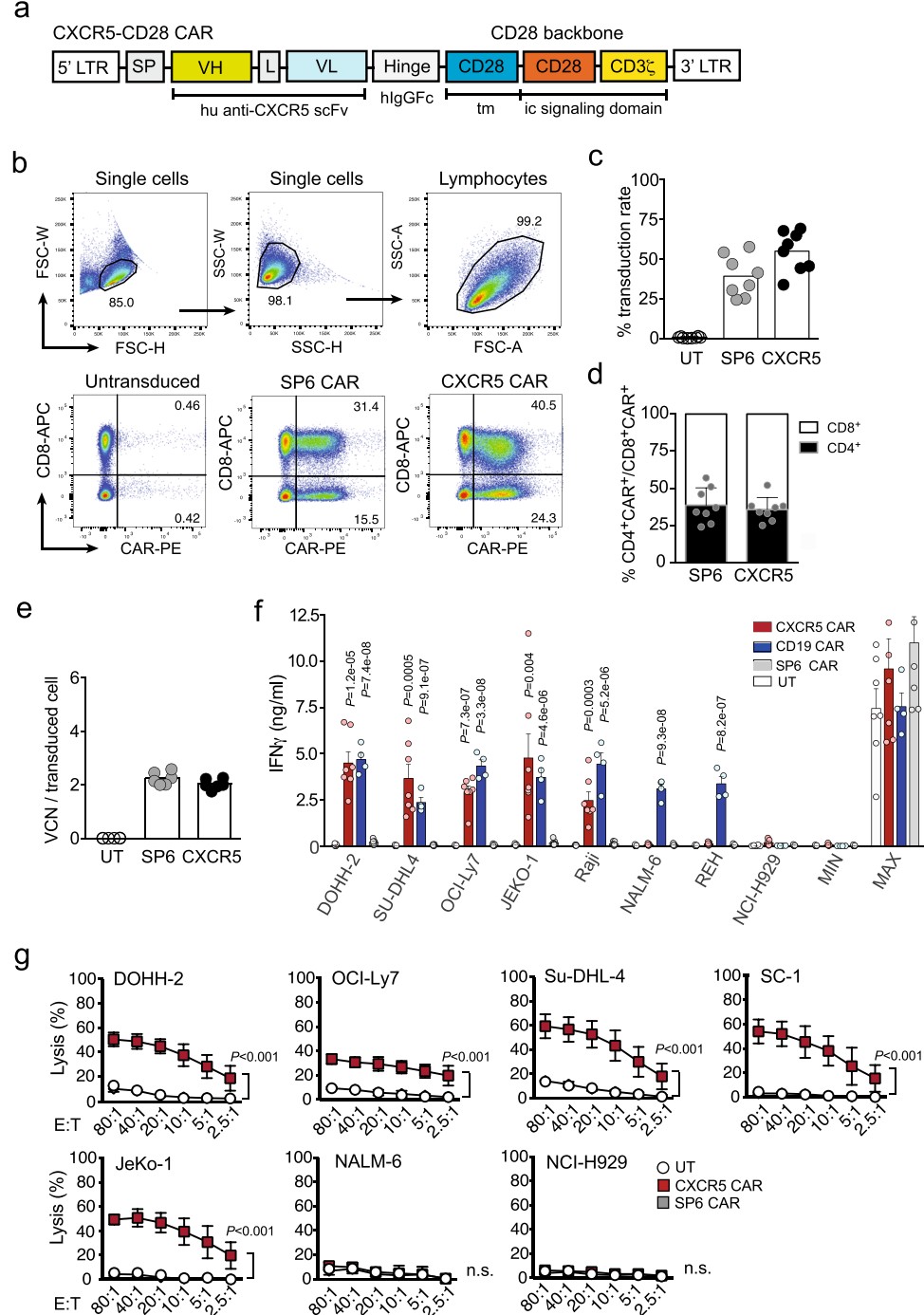

**Fig. 2 The CXCR5 CAR construct is efficiently expressed and confers T cells with selective effector functions against CXCR5-expressing tumor cell lines. a** Schematic representation of the anti-human CXCR5-CD28 CAR construct. SP signal peptide, VH variable heavy chain, L Whitlow linker, VL variable light chain, tm transmembrane region, ic intracellular, LTR long terminal repeat, hIgGFc human IgG1 Ch2CH3 constant region. **b** CAR surface expression on human T cells transduced with CXCR5 or SP6 (control) CARs, or left untransduced (UT), analyzed on day 11 after transduction by staining with an anti-IgG antibody that detects the common CAR extracellular spacer domain. Percentages of CD8+CAR+ and CD8-CAR+-T cells are indicated. **c** Graphs represent the percentage of CXCR5 and SP6 CAR-expressing cells and **d** the proportion of CAR-expressing CD4+ (solid bars) relative to CD8+ (open bars) T lymphocytes. **e** Viral copy number (VCN) per transduced cell was determined in CXCR5 CAR- (black points) and SP6 CAR-transduced (gray points) or UT (open circles) T cells. Representative data in (**b**) and mean ± SEM values shown as bars in (**c**) and (**d**) were generated by $n = 6$ independent experiments and $n = 8$ individual donors. **e** Data of $n = 6$ individual donors. **f** UT, SP6, CXCR5 or CD19 CAR-transduced T cells ($5 \times 10^4$) were cocultured in a 1:1 ratio with B-NHL cell lines (DOHH-2, SU-DHL4, OCI-Ly7, JeKo-1), Raji (Burkitt lymphoma), B-ALL (NALM-6, REH), and MM (NCI-H929) cell lines. IFNγ content in the supernatant was determined by ELISA. Max, CAR-T cells stimulated with PMA/Ionomycin; Min, CAR-T cells only. Bars represent mean ± SEM; $P$ values are calculated for CXCR5, or CD19 CAR-T cells compared to SP6 by an unpaired two-tailed multiple $t$ test. **g** CAR-T cell-mediated tumor cell killing was analyzed in a standard $^{51}$Cr release assay. $^{51}$Cr-labeled B-NHL cell lines (as indicated) were cocultured with UT, SP6, and CXCR5 CAR-transduced T cells at the indicated effector: target (E:T) ratios for 4 h. CXCR5-negative cell lines Nalm-6 and NCI-H929 served as controls. One out of two independent experiments is shown with $n = 3$ donors. Values displayed are the mean ± SEM with $P < 0.001$ by Kruskal–Wallis test. Source data are provided as a Source Data file.

can be produced in clinically meaningful numbers despite some T cell fratricide leading to the elimination of CXCR5+ T cells.

To determine the antigen-specificity of CXCR5 CAR-T cells, we co-cultured them together with B-NHL cell lines and compared CXCR5 CAR-T with CD19 CAR-T cell reactivity. The CAR-T cells showed similar transduction rates (Fig. 2c, Supplementary Fig. 7a, b) and both reacted strongly towards all six mature B-NHL cell lines (DOHH-2, SU-DHL4, OCI-Ly7, MEC-1, SC-1, JeKo-1) with profound IFNγ secretion. Confirming the specificity of the CXCR5 CAR, the CXCR5−CD19+ B-ALL cell lines (NALM-6, REH) were not recognized by the CXCR5 CAR, but by the CD19 specific CAR-T cells. Neither of the CAR-T cells reacted against the CXCR5−CD19− NCI-H929 line (Fig. 2f). Notably, similar antigen-specific IFNγ secretion was observed for both, the CXCR5 CAR-T as well as the CD19 CAR-T cells, although CD19 expression levels (Supplementary Fig. 7c) were on average 5–10-times higher than for CXCR5 (Fig. 1b) on all tumor cell lines.

In addition, we assessed the cytolytic capacity of CXCR5 CAR-T cells against the DOHH-2, OCI-Ly7, SU-DHL-4, SC-1, and JeKo-1 B-NHL cell lines and against a B-ALL and a MM cell line (NALM-6 and NCI-H929, respectively). In agreement with the IFNγ response, we observed robust cytolytic activity against all B-NHL cell lines. The cytolytic activity was antigen-specific as NALM-6 and NCI-H929 cells were ignored by CXCR5 CAR-T cells, even at high effector-to-target ratios (Fig. 2g).

**Anti-CXCR5 CAR-T cells mediate targeting of both primary lymphoma B cells and lymphoma-supporting Tfh cells.** The CXCR5 CAR not only endowed T cells with antigen recognition on lymphoma cell lines, but also on a large number of primary patient-derived FL, B-CLL, MCL, and MZL samples (Fig. 3a–d). Overall, CXCR5 CAR-T cells reacted stronger towards primary FL, CLL, and MCL samples with respect to IFNγ secretion compared to CD19 CAR-T cells (Fig. 3a–c). In contrast, MZL expressed little CXCR5 except for one sample, and consequently were hardly recognized by CXCR5 CAR-T cells (Figs. 1c, d, 3d). CD19 CAR-T cells on the other hand released much higher amounts of IFNγ upon MZL B cell encounter (Fig. 3d).

Tfh cells (CD4+PD1+CXCR5+) can exhibit a tumor-supporting role in human B cell malignancies, foremost in FL and CLL[18]. To prove CXCR5 CAR reactivity against Tfh cells, we co-cultured CXCR5 CAR-T cells with sorted CD4+CXCR5+ T helper cells as a surrogate for intratumoral Tfh cells. These cells elicited a robust IFNγ release by CXCR5 CAR-T cells, but not by CD19 CAR-T cells (Fig. 3e), indicating that Tfh cells are potential targets of CXCR5 CAR-T cells.

Next, we assessed whether the CXCR5 CAR also mediates the superior killing of FL, CLL, and MCL lymphoma cells in

comparison to the CD19 CAR. Therefore, we analyzed the frequencies of FL, CLL, and MCL cells and T cells in 48-h co-cultures by flow cytometry (Fig. 3f–h; Supplementary Fig. 8a–e). CXCR5 CAR-T cells killed FL, CLL, and MCL cells more effectively than CD19 CAR-T cells, demonstrating superior anti-lymphoma cell activity. Only the MCL cell line JeKo-1 was depleted by CXCR5 and CD19 CAR-T cells at a similar rate (Fig. 3f–h). Differences in the homogeneity and height of CXCR5 and CD19 expression on the primary lymphoma cells may play a role in this observation.

Another difference is that CD4+PD1+CXCR5+ Tfh cells within FL and CLL samples were completely depleted by the CXCR5 CAR-T cells whereas the CD19 CAR-T cells had no effect on Tfh cell numbers (Fig. 3i; Supplementary Fig. 8d, e). Overall, our results emphasize that CXCR5 is an attractive alternative target for lymphoma entities that cannot effectively be controlled by CD19 CARs. Secondly, Tfh cells as part of the tumor microenvironment (TME) are concomitantly destroyed by CXCR5 CAR-T but not by CD19 CAR-T cells.

**CXCR5 shows a highly restricted expression profile across human tissues.** To rule out that CXCR5 CAR-T cells cross-react with healthy human tissues, we analyzed a large panel of primary human, non-lymphoid cells, and cell lines by flow cytometry for CXCR5 expression. Human primary umbilical vein endothelial cells, and umbilical artery endothelial cells, astrocytes, perineurial cells and neurons from the CNS, and epithelial cells from the colon, cervix, and bladder did not express CXCR5 on their surfaces (Fig. 4a). Human cancer cell lines that originated from the liver, colon, and kidney were also negative for CXCR5 (Fig. 4b). To rule out that CXCR5 expression is enforced under inflammatory conditions, we stimulated a number of primary human cells with IFNγ. This treatment caused upregulation of MHC class I molecules, but surface deposition of CXCR5 protein was not recorded (Fig. 4a), indicating that CXCR5 expression is not upregulated upon IFNγ-induced inflammation.

In accordance with the lack of CXCR5 expression, co-incubation of CXCR5, CD19, SP6 CAR-transduced T cells with various primary cell types either without (Fig. 4c) or with an inflammatory stimulus (Fig. 4d) did not induce cross-reactivities or on-target T cell activation. Collectively, the CXCR5 CAR-T cells exhibited no cross-reactivity or on-target/off-tumor effects against a wide range of human cell lines or primary cells.

**CXCR5 CAR-T cells exhibit strong anti-lymphoma activity in NSG mice.** To show that the human CXCR5 CAR-T cells can mount an efficient in vivo antitumor activity, NSG mice were xenotransplanted with $6 \times 10^5$ JeKo-1 cells stably expressing

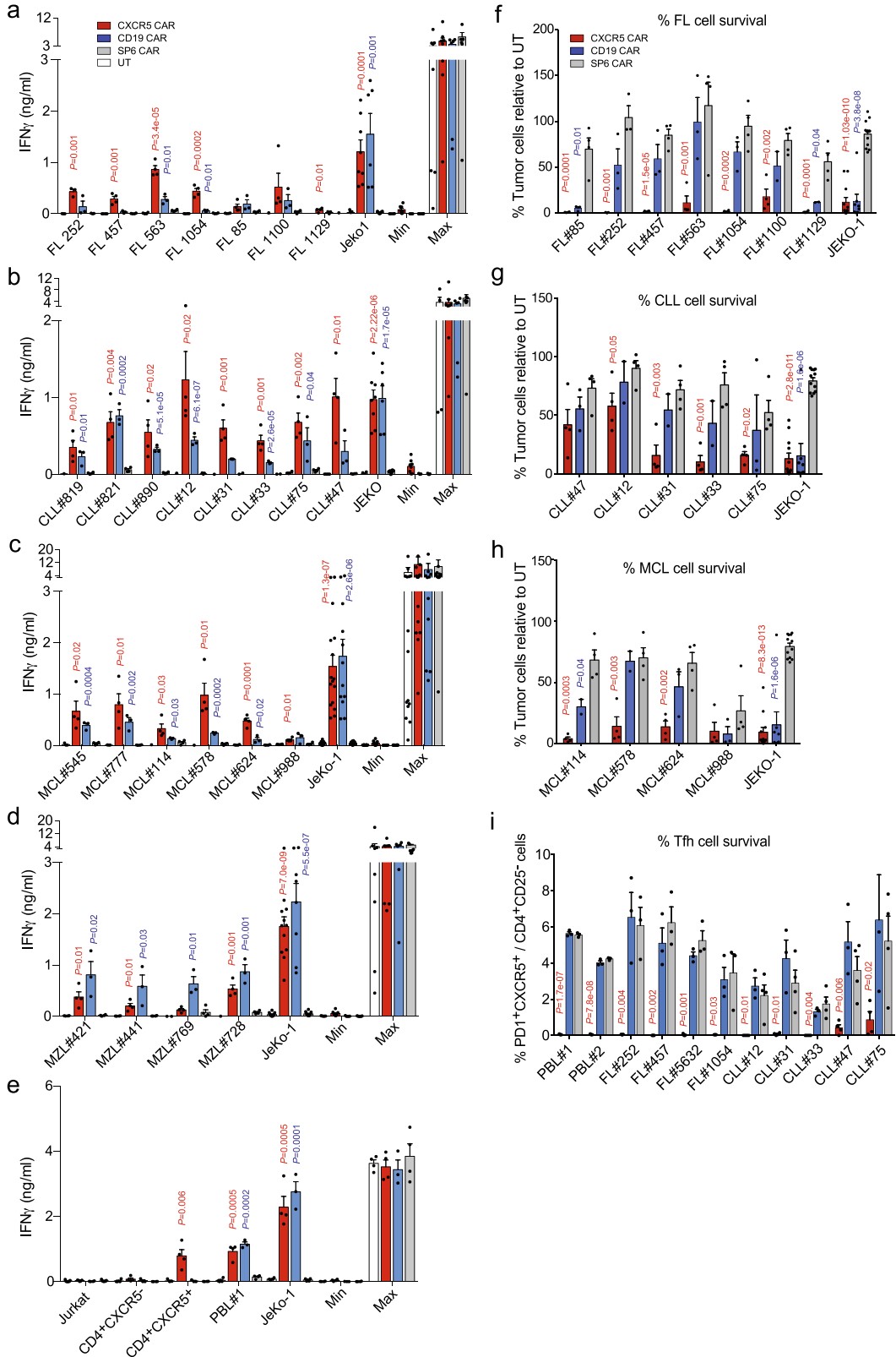

luciferase and GFP (Fluc/GFP)[31]. Six days later, tumor growth was confirmed by bioluminescence imaging (BLI), and $3 \times 10^6$ CXCR5 or SP6 CAR-T cells were administered i.v. (=day 0 after T cell transfer) (Fig. 5a). In mice treated with control SP6 CAR-T cells, successive luminescence measurements indicated progressive tumor growth. Application of CXCR5 CAR-T cells abrogated disease progression almost completely throughout the

observation time of 23 days (Fig. 5b, c). Overall, the CXCR5 CAR endowed transduced T cells with a robust in vivo anti-tumor activity.

Next, we compared the anti-tumor activity of the CXCR5 and CD19 CAR-T cells. In two independent experiments, NSG mice were xenotransplanted with JeKo-1 cells. At day 5, $3 \times 10^6$ CAR-T cells were administered i.v. (Fig. 6a). Application of CXCR5 or

**Fig. 3 The CXCR5 CAR confers T cells with selective effector functions against CXCR5-expressing primary B-NHL cells and Tfh cells.** UT, SP6, CXCR5 or CD19 CAR-transduced T cells ($5 \times 10^4$) were cocultured in a 1:1 ratio for 12–24 h with patient-derived primary **a** FL cells, **b** CLL cells, **c** MCL cells, **d** MZL cells, and **e** primary sorted T cell subpopulations and PBLs. Data were pooled from 2 to 3 experiments. JeKo-1 cells served as positive controls. IFNγ in the supernatant was determined by ELISA. Max, CAR-T cells and PMA/Ionomycin; Min, CAR-T cells only. **a–e** Bars represent $n = 4$ (UT, CXCR5 and SP6 CAR) or $n = 3$ (CD19 CAR) individual donors, with the exceptions: FL#252 in (**a**) $n = 3$ for UT, CXCR5 and SP6 CAR, CLL#819, #821 and #890 in (**b**) $n = 1$ for UT. Controls JeKo-1, Min and Max represent $n = 8$ (UT, CXCR5 and SP6 CAR) or $n = 6$ (CD19 CAR) individual donors in (**a**, **b** and **d**), with the exceptions: UT and JeKo-1 in (**d**) were $n = 5$ for CD19 CAR, Max in (**d**) was $n = 7$ for SP6 CAR. Controls in (**c**) represent $n = 12$ measurements of eight individual donors (UT, CXCR5, and SP6 CAR) and $n = 9$ measurements of 6 individual donors (CD19 CAR). Controls in (**e**) represent $n = 4$ (UT, CXCR5, and SP6 CAR) or $n = 3$ (CD19 CAR) individual donors. **f–i** 48-h cocultures of primary B-NHL samples (as indicated) with CXCR5, CD19, or SP6 CAR-T cells were analyzed by flow cytometry (gating strategies are shown in Supplementary Fig. 8). Bars show **f–h** the percentages of tumors cells normalized to cocultures of untransduced T cells with tumor samples and **i** the percentages of CXCR5+PD1+ T cells within the CD4+CD25− population. **f–h** $n = 4$ (UT, CXCR5 and SP6 CAR) or $n = 3$ (CD19 CAR) individual donors, with the exceptions: FL#252 in (**f**) was $n = 3$ for CXCR5 and SP6 CAR, FL#1100 and #1129 in (**f**), CLL#12, #31 and #33 in (**g**) and MCL#114 and #578 in (**h**) were $n = 2$ for CD19 CAR. **i** $n = 3$ individual donors, with the exception: CLL#12, #31, #33, #47, #75 were $n = 4$ for CXCR5 and SP6 CAR. Bars represent mean ± SEM; $P$ values are calculated for CXCR5 (red), or CD19 CAR-T cells (blue) compared to SP6 by an unpaired two-tailed multiple $t$ test. Source data are provided as a Source Data file.

CD19 CAR-T cells conferred robust antitumor effects over 16–18 days while in mice treated with control SP6 CAR-T cells increasing luminescence signals indicated disease progression (Fig. 6b, c and Supplementary Fig. 9a–c). Notably, JeKo-1 tumors eventually recurred in mice treated with either CXCR5 or CD19 CAR-T cells beyond 16–20 days. Residual JeKo-1 cells recovered from BM and spleen on day 19 or 20 showed conserved CXCR5 expression in all three CAR-T cell groups, suggesting that CXCR5 CAR treatment did not select for antigen loss variants (Fig. 6d and Supplementary Fig. 9d). We also examined the possibility that shedding of the CXCR5 N-terminus covering the cognate epitope might interfere with CXCR5 CAR-T cell activity as it has been investigated for the BCMA antigen[32,33]. A 21-aa peptide which represents the epitope recognized by the anti-CXCR5 CAR, and a control peptide (17 aa) covering part of the CXCR5 first extracellular loop were synthesized. Peptides were added at increasing concentrations to a co-culture of JeKo-1 tumor cells with CXCR5 or CD19 CAR-T cells. IFNγ release by the activated CAR-T cells was determined after 24 h and revealed no inhibition of anti-CXCR5 CAR-T cell activity in the presence of a soluble CXCR5 N-terminal peptide at physiological concentrations (Supplementary Fig. 10).

To further explore why tumor relapse occurs in the NSG mouse model, we analyzed the persistence of CXCR5 CAR-T cells in vivo at the time of JeKo-1 outgrowth. NSG mice were xenotransplanted with JeKo-1 cells and received CXCR5 or control CAR-T cells ($3 \times 10^6$ transduced CAR+ T cells) six days later. Mice treated with the SP6 CAR-T cells developed progressive disease between 4 and 12 days after CAR-T cell infusion, whereas animals treated with the CXCR5 CAR-T cells showed a robust antitumor effect (Supplementary Fig. 11a, b). Over the course of the experiment, 4–5 mice per time point were analyzed for tumor load, CXCR5 expression of tumor cells, and total CD4 and CD8 T cell numbers in BM and spleen. Tumor load in the BM was profoundly reduced in mice treated with CXCR5 CAR-T cells compared with SP6 CAR-T cells at day 8, and in BM and spleen at day 13 as well (Supplementary Fig. 11c). Residual JeKo-1 cells recovered from BM and spleen had a conserved expression of CXCR5 in both CAR-T cell groups (Supplementary Fig. 11d), underlining that tumor recurrence was not a result of the selection for CXCR5 antigen loss variants. Consistent with an expansion of antigen reactive CAR-T cells, at day 8 higher numbers of CD4+ T cells in BM and CD8+ T cells in the spleen were detected. On day 13 SP6 and CXCR5 CAR-T cells were detected at similar levels in BM as well as in the spleen (Supplementary Fig. 11e). Thus, at primary sites of tumor cell homing CXCR5 CAR-T cells were specifically amplified and were detectable up to 13 days after adoptive transfer. The exhaustion

markers PD-1, LAG-3, and TIM-3 were not differentially upregulated over 13 days on BM or splenic localized CD4+ and CD8+ CXCR5 CAR-T compared to SP6 CAR-T cells (Supplementary Fig. 11f–h), thus exhaustion seems to be unlikely involved in loss of tumor control.

**CXCR5 CAR-T cells maintain their functionality upon recursive antigen exposure.** To assess whether chronic CAR-mediated activation could lead to T cell dysfunction, we employed a co-culture system whereby every 72 h CXCR5 CAR-T or CD19 CAR-T cells were recursively transferred to culture dishes seeded with JeKo-1 cells, adjusting for a constant viable 1:1 CAR-T cell: tumor cell ratio[31,34,35]. We performed five repetitive stimulation rounds and determined T cell functionality and exhaustion (Fig. 7a). CAR-T cells were utilized on day 14 after the start of their cultivation period. During recursive activation cycles, antitumor cytolytic activity, IFNγ secretion, and the proliferative capacity of both the CXCR5 as well as the CD19 CAR-T cells were similarly maintained (Fig. 7b). Repetitive stimulation led to an enrichment of the CXCR5 and CD19 CD4+ and CD8+ CAR-T cell population to almost 100% (Fig. 7c). The expression of PD-1, LAG-3, and TIM-3 on CD8+CAR+-T cells was not altered during repetitive stimulation (Fig. 7d). CD4+CAR+-T cells behaved similarly, but they displayed a higher expression of PD-1 and a lower expression of LAG-3 compared with CD8+CAR+-T cells (Fig. 7d).

To demonstrate that other constructs or different culture conditions can induce T cell dysfunction, we altered the CAR-T cell: tumor cell ratio (1:2) in the same coculture system. Moreover, we introduced a first-generation CXCR5 CAR-T cell construct lacking the CD28 co-stimulatory domain. As expected, although effective in killing, the first-generation CXCR5 CAR-T cells did not survive beyond the second round of stimulation. Also the altered CAR-T cell: tumor cell ratio led to the dysfunction of CXCR5 CAR as well as CD19 CAR-T cells from the third round of stimulation on (Supplementary Fig. 12a) accompanied by enhanced PD-1 expression on CD8+ and CD4+ CAR-T cells (Supplementary Fig. 12b).

Overall, these data demonstrate that the functional capacities of CXCR5 CAR-T cells can be maintained over an extended stimulation period in a similar manner as CD19 CAR-T cells.

**Anti-murine CXCR5 CAR-T cells gain access to tumor cells in the B cell follicles and exert potent anti-lymphoma activity.** CXCR5 is expressed in human and mouse tissues with an overlapping pattern. Thus, we engineered an anti-mouse CXCR5 CAR aiming at overcoming the major limitations of the xenotransplantation NSG mouse model. The syngeneic immunocompetent mouse

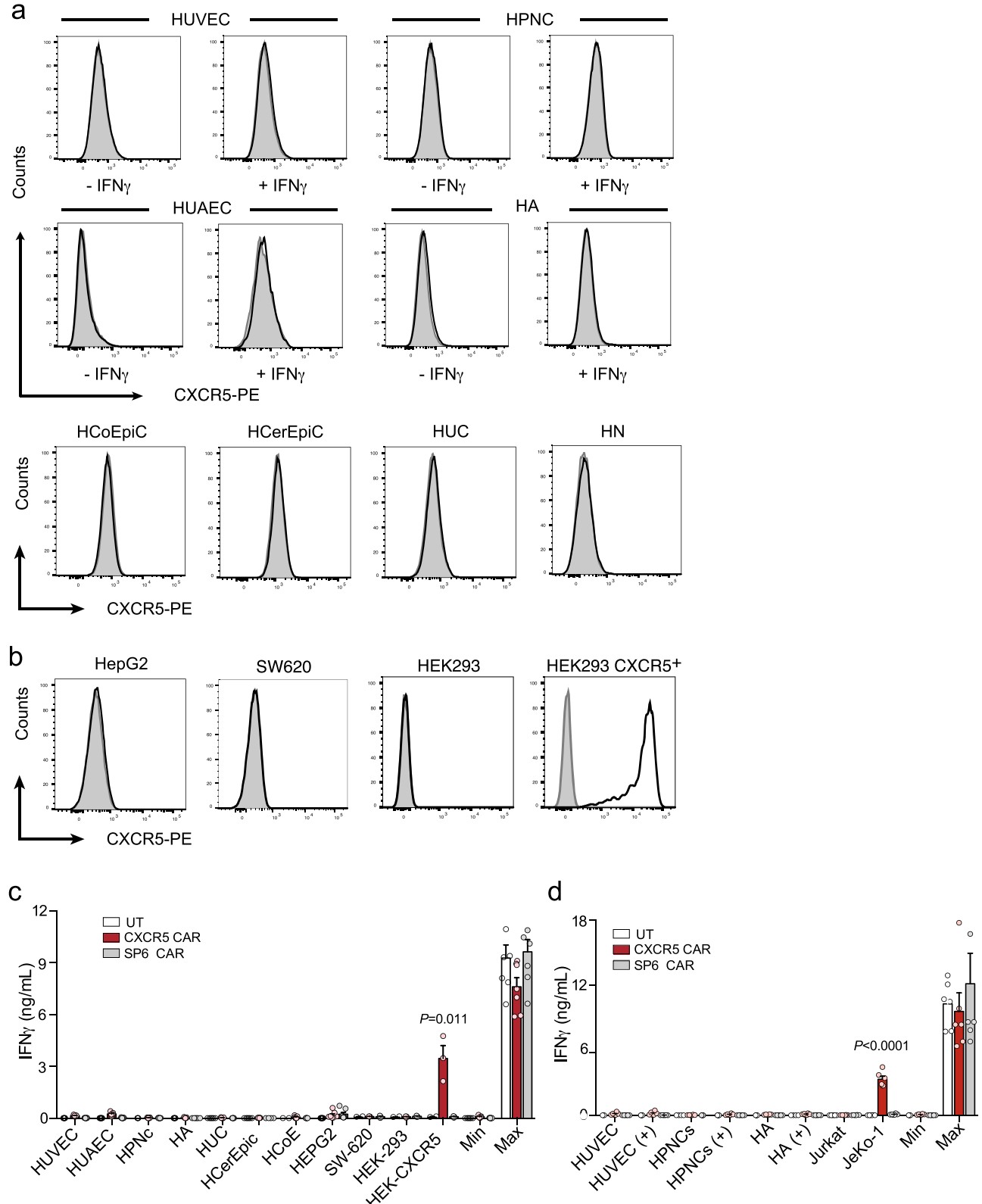

model allows us to study anti-lymphoma activity in an organized lymphoid microenvironment including B and T cell areas and to record on-target/off-tumor toxicity. The binding moiety of the mouse CXCR5 (mCXCR5) CAR is based on the rat anti-mouse CXCR5 mAb (2G8), reactive against the N-terminal domain of mouse CXCR5 (Supplementary Fig. 13a)[20]. We designed the anti-mouse CXCR5 CAR (referred to as mCXCR5 CAR) similar to the anti-

human CXCR5 CAR, however, all modules were of murine origin (Fig. 8a). Murine splenic T cells were transduced with the mCXCR5 CAR or GFP as negative control. CAR surface expression was in the range of >40% in murine T cells (Fig. 8b). The mCXCR5 CAR-T cells reacted specifically against primary murine CXCR5[+/+] *Eμ-Tcl1* leukemia cells, splenic B220[+] B cells, and CXCR5-transduced BW5147 cells as indicated by a high release of mIFNγ and mIL2 (Fig. 8c),

**Fig. 4 Primary human non-lymphoid cells derived from healthy tissues and cancer cell lines do not activate anti-CXCR5 CAR-T cells.** CXCR5 surface expression on (**a**) primary human cells (HUVEC, human umbilical vein endothelial cells; HUAEC, human umbilical artery endothelial cells; HPNC, human perineurial cells; HA, human astrocytes; HCoEpiC, human colonic epithelial cells; HCerEpic, human cervical epithelium cells; HUC, human urothelial cells; HN, human neurons) and on (**b**) cancer cell lines (HepG2, hepatocellular carcinoma; SW620, colon carcinoma; HEK293, embryonic kidney epithelial cells). Selected primary cells were stimulated for 24 h with (+) or without (−) IFNγ (25 ng/ml). CXCR5-transduced HEK293 cells served as a positive control. **c**, **d** UT, SP6, or CXCR5 CAR-transduced T cells were cocultured for 22 h with human primary cells and cell lines pretreated with or without IFNγ (25 ng/ml). The JeKo-1 cell line served as a positive control. IFNγ release was quantified in supernatants by ELISA. Max, CAR-T cells and PMA/Ionomycin; Min, CAR-T cells only. **a**, **b** Representative data of $n = 2$ independent experiments. **c**, **d** $n = 2$ independent experiments with $n = 7$ (**c**) or $n = 6$ (**d**) individual donors, except for SW620, HEK293, and HEK CXCR5+ in (**c**), which show $n = 3$ individual donors. Bars represent mean ± SEM; $P$ values are calculated for CXCR5 CAR-T cells compared to SP6 CAR-T cells by an unpaired two-tailed Student's $t$ test. Source data are provided as a Source Data file.

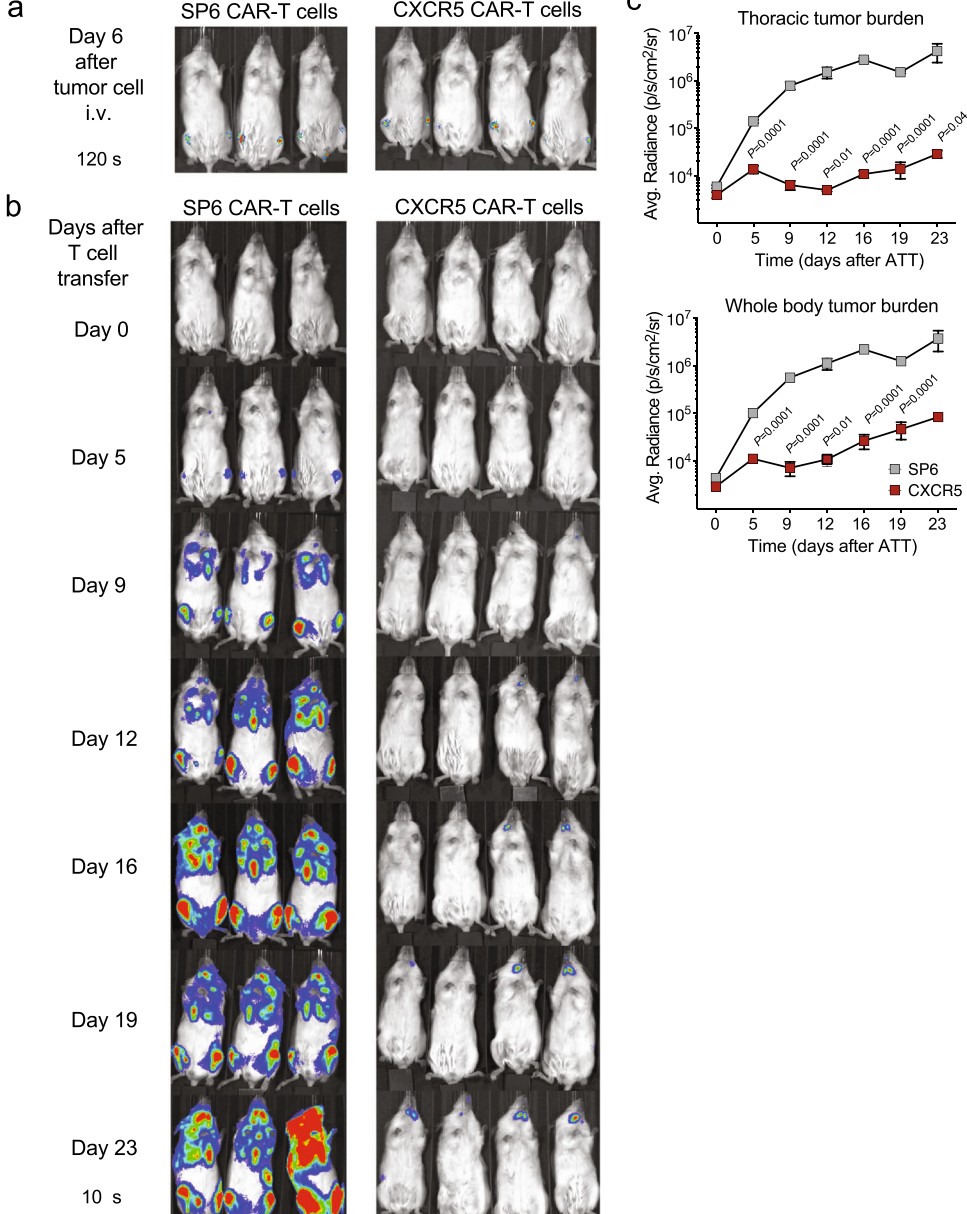

**Fig. 5 CXCR5 CAR-T cells exhibit efficient antitumor killing activity in vivo against B-NHL xenograft. a** Engraftment of $6 \times 10^5$ i.v. transplanted JeKo-1 cells (MCL; GFP+luc+) in NSG mice. Six days later, tumor growth was visualized by IVIS imaging (exposure: 120 s). **b** $3 \times 10^6$ CAR-transduced T cells (total T cells: 5–6 × 10⁶) were transferred into tumor-bearing mice on day 6 (=day 0 after T cell transfer). Images of serial IVIS-exposures of mice treated with SP6 ($n = 3$) or CXCR5 CAR-T cells ($n = 4$) are shown (exposure: 10 s instead of 120 s to allow better comparisons between day 0 and 23). **c** Means of the signal intensities obtained of thoracic or from the entire body tumor burden are plotted for each group over time. Values displayed as means ± SEM, $P$ values were calculated by unpaired two-tailed Student's $t$ test comparing animals per group at the same time point. Source data are provided as a Source Data file.

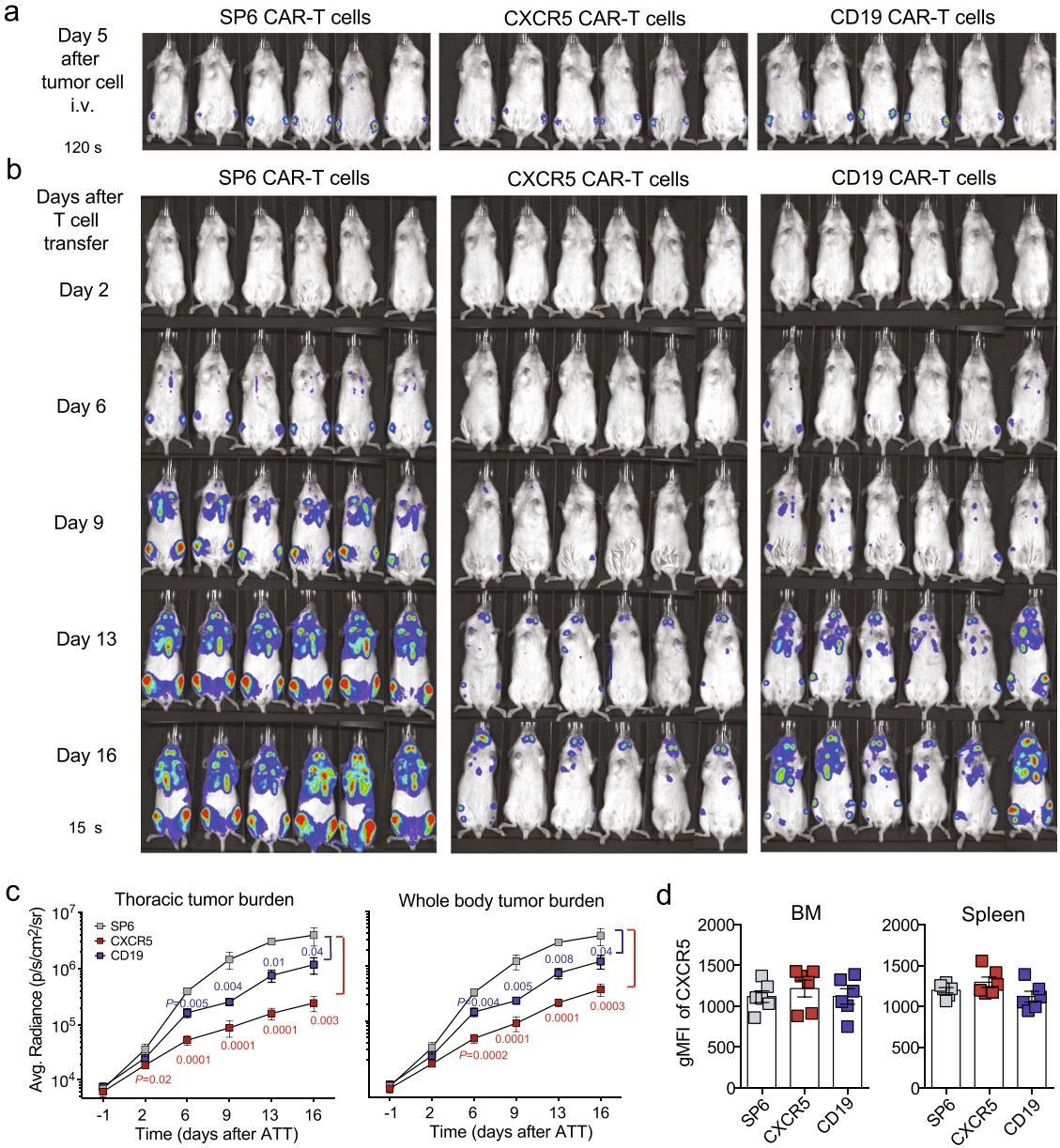

**Fig. 6 CXCR5 CAR-T cells exhibit a similar antitumor cytolytic activity against B-NHL xenografts as CD19 CAR-T cells. a** Engraftment of $6 \times 10^5$ i.v. transplanted JeKo-1 cells (MCL; GFP+luc+) in NSG mice. Five days later, tumor growth was visualized by IVIS imaging (exposure: 120 s). **b** Tumor-bearing mice received $3 \times 10^6$ CAR-T cells (total T cells: $6-7 \times 10^6$) on day 7 and were serially imaged (exposure: 15 s) from day 2 to 16 post T cell infusion (SP6 CAR: $n = 6$; CXCR5 CAR: $n = 6$; CD19 CAR: $n = 6$). **c** Means of the signal intensities obtained from of thoracic or entire body tumor burden are plotted for each group over time. Values displayed as means ± SEM, P values were calculated by unpaired two-tailed Student's $t$ test comparing animals of the CXCR5 CAR (day 2–16) or CD19 CAR group (day 6–12) compared to the SP6 group at same the time point ($n = 6$, per group). **d** CXCR5 expression on JeKo-1 tumor cells derived from BM or spleen at day 19 or 20 after CAR-T cell transfer analyzed in the GFP+ population. Bars represent geometric mean fluorescence intensities (gMFI) of CXCR5 ± SEM of $n = 6$ mice per group. Source data are provided as a Source Data file.

whereas CXCR5$^{-/-}$ Eµ-Tcl1 cells and non-transduced BW5147 cells failed to activate mCXCR5 CAR-T cells.

Next, we adoptively transferred $2 \times 10^6$ mCXCR5 CAR- and GFP-transduced T cells into sublethally irradiated B6 mice. Seven, 14, and 20 days later, the numbers of endogenous CXCR5$^+$B220$^+$ B cells (Fig. 8d) as well as CXCR5$^+$CD4$^+$ T cells (Fig. 8e) were profoundly reduced in the mCXCR5 CAR-T cell-treated group compared to controls, suggesting a specific activity of mCXCR5 CAR-T cells against mature B and a subset of T helper cells. B and T cell depletion lasted up to 20 days after mCXCR5 CAR-T cell transfer. Mice from both groups had no inflammatory mono-nuclear cell infiltrates in liver, lung, and colon at day 7

(Supplementary Fig. 13b), and serum markers indicative of major organ damage in CAR-T cell recipient mice were not altered compared to naive mice over the whole course of the experiment (Supplementary Fig. 13c). Collectively, mCXCR5 CAR-T cells facilitated antigen-specific recognition of benign and malignant B cells in vitro accompanied by specific B and selective T cell depletion without off-target reactivity in vivo.

Because CXCR5 CAR-T cells also target a subset of CD4$^+$ CXCR5$^+$ T cells, including the CD4$^+$CXCR5$^+$ Tfh cells (Figs. 3i, 8e), we asked whether CXCR5 CAR-T cell recipient mice would be severely impaired in mounting a regular T cell immune response. Here, we transferred $2 \times 10^6$ mCXCR5 or mSP6 CAR-T cells into

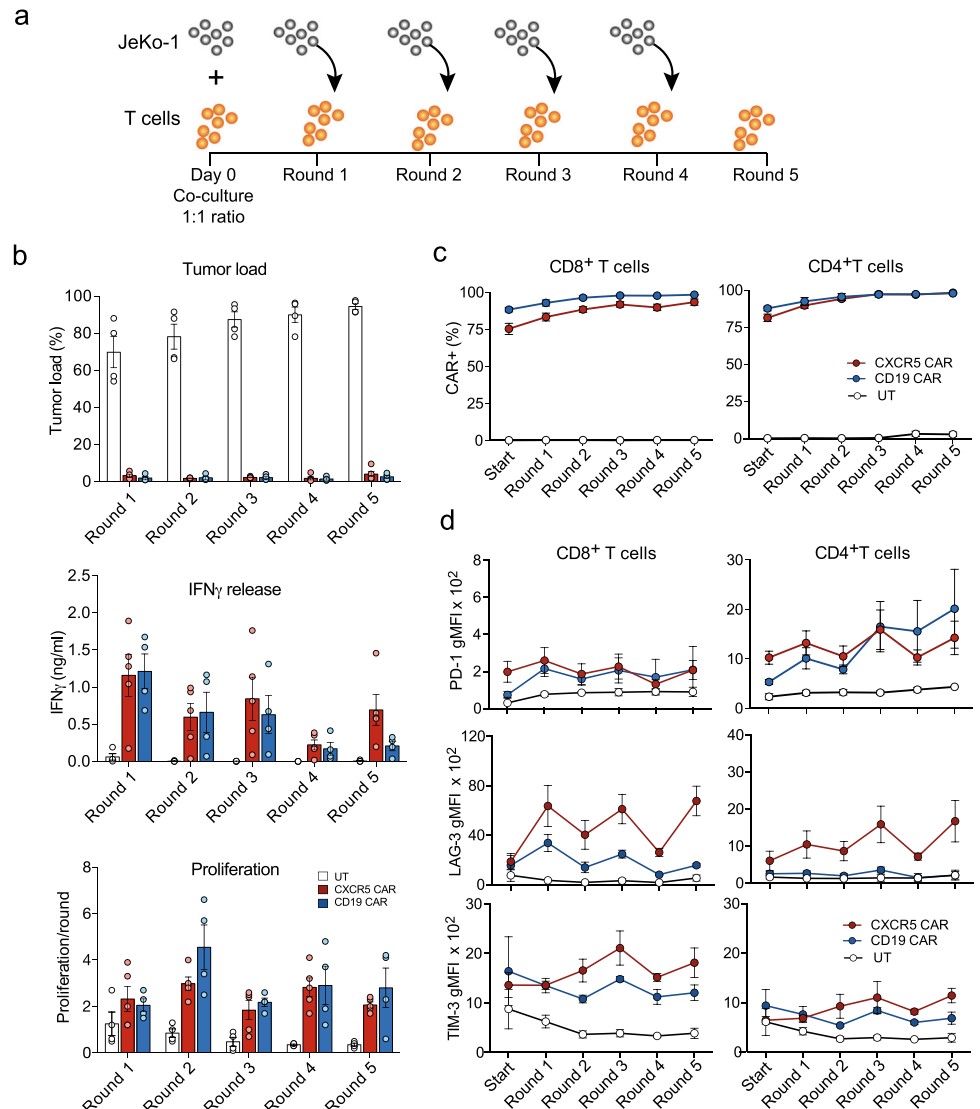

**Fig. 7 CXCR5 CAR-T cells exhibit extended effector function, viability, and proliferation during recursive antigen exposure in vitro. a**, **b** CXCR5 and CD19 CAR-T cells were cultivated for 10–14 days before they were cocultured with JeKo-1 cells in two different E:T ratios: **b** 1:1 or (Supplementary Fig. 12) 1:2. Every 72 h, a repetitive transfer of CAR-T cells into wells with fresh target cells was conducted for a total of five rounds re-establishing the initial E:T ratios. Untransduced (UT) T cells served as controls. For the experiments with the 1:2 ratio, also a first gen. CXCR5 CAR was included for comparison (Supplementary Fig. 12). **b** Three parameters were determined at the end of each round: (i) tumor load (7-AAD[-]/CXCR5[+]CD3[-]) as determined by flow cytometry, (ii) IFNγ release as determined by ELISA, (iii) proliferation of T cells per round (*n* cells end/*n* cells start). **c** The percentage of CAR[+] T cells and **d** expression of PD-1, LAG-3, and TIM-3 on CAR[+] T cells. Isotype geometric mean fluorescence intensities (gMFI) were subtracted from the gMFIs of the specific stainings. Data in (**b–c**) are pooled from *n* = 3 independent experiments and represent *n* = 4 (UT, CD19 CAR) or *n* = 5 (CXCR5 CAR) individual donors; Bars represent means, errors are SEM. Source data are provided as a Source Data file.

irradiated mice (Fig. 8f). On day 19, the continued presence of mCXCR5 CAR-T cells and B cell depletion in the host as a readout for their efficacy were confirmed (Fig. 8g, h). Next, animals were immunized with SV40 large T antigen (Tag[+]) expressing 16.113 tumor cells that normally elicit a strong T cell response in immunocompetent animals. Eight days later, we detected antigen-specific, Kb-Tag IV dextramer[+], CD8[+] T cells in both the mCXCR5 and the mSP6 CAR-T cell group at similar frequencies (Fig. 8i). This result proves that mice treated with anti-CXCR5 CAR-T cells are still able to mount antigen-specific T cell responses.

Previously, we showed that upon adoptive transfer of murine *Eμ-Tcl1* leukemia cells into healthy recipients, leukemia cells home in a strictly CXCR5-dependent manner into B cell follicles where they get in close contact to follicular dendritic cells (FDCs), the most important follicular stromal cell network. Access to and

crosstalk with FDCs supported leukemia cell survival and expansion[27]. Hence, we addressed the crucial question if CXCR5 CAR-T cells that lack CXCR5 expression can enter B cell follicles and efficiently kill follicular leukemic cells in a short-term CAR-T cell transfer model. We transferred $2 \times 10^6$ *Eμ-Tcl1* tumor cells i.v. into congenic Ly5.1 B6 mice. Eight days later, recipient mice were sublethally irradiated and $2 \times 10^7$ CXCR5 CAR-T cells were administered i.v. The sublethal irradiation mimics pre-conditioning of leukemia patients prior to CAR-T cell treatment, but does not eradicate leukemia cells. One and five days after CAR-T cell transfer, the frequencies and anatomic localization of normal B cells, leukemic cells, and mCXCR5 CAR-T cells in the spleen were determined by immune histology and by flow cytometry (Fig. 9a). As depicted in Fig. 9b and Supplementary Fig. 14, T and B cell compartments were still maintained 2 days

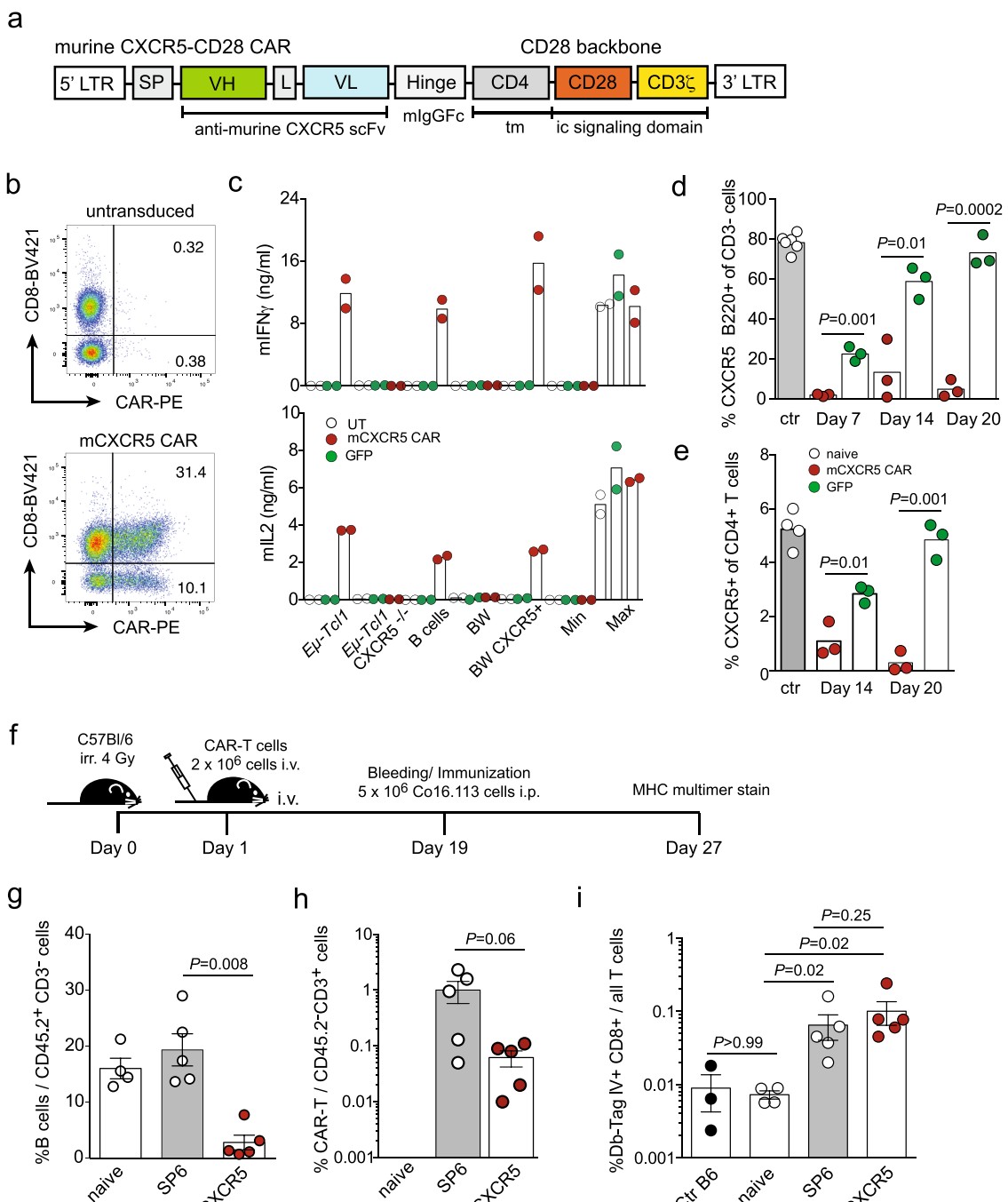

after irradiation and 24 h after CAR-T cell treatment. The FDC network (CD21/CD35+) within the B cell follicles (Fig. 9b, Supplementary Fig. 14a) and the gp38+ fibroblastic reticular cell (FRC) network within the T cell zones (Supplementary Fig. 14b) were still present, although stromal networks as well as B cell follicles were reduced in size compared to the controls. Control animals received leukemia cells but no irradiation or further CAR-T cell treatment. Most importantly, 24 h after mCXCR5 CAR-T cell transfer, CAR-T cells closely intermingled with leukemic B cells within the remaining B cell areas, defined by the presence of FDC networks. Five days after CAR-T cell transfer, efficient leukemic cell reduction and a profound expansion of CAR-T cells were observed (Fig. 9b, Supplementary Fig. 10a). B cell areas were dissolved, FDC networks essentially disappeared, and CAR-T cells predominantly expanded within the partially conserved gp38 networks (Supplementary Fig. 14b). Flow

cytometry data confirmed these results and showed significant reduction of percentages and total numbers of leukemic cells (Fig. 9c), percentages of endogenous B cells (Fig. 9d), and a significant increase in percentages and total numbers of CAR-T cells in the spleen at day 5 compared to 24 h after CAR-T cell transfer (Fig. 9e).

To study the anti-leukemia efficacy in a therapeutic approach, we transferred $2 \times 10^6$ $E\mu$-$Tcl1$ cells, followed by a sublethal irradiation on day 7 and the administration of $2 \times 10^6$ CXCR5 or SP6 CAR-T cells on day 8 (Fig. 9f). At day 21 after leukemia cell transfer, we found a strong depletion of $E\mu$-$Tcl1$ cells, endogenous B cells, and CXCR5+CD4+ T cells in the spleens of the mCXCR5 CAR group compared to the control (Fig. 9g). A similar ratio of transferred CD4+ and CD8+ T cells was found in both groups, however, the proportion of CAR-positive T cells among all transplanted T cells was higher in the SP6 CAR compared to the

**Fig. 8 Anti-murine CXCR5 CAR T cells selectively delete CXCR5$^+$ benign lymphocyte subsets without harming the host's T cell immunocompetence.** **a** Schematic representation of the anti-murine CXCR5-CD28 CAR (mCXCR5 CAR). SP signal peptide, VH variable heavy chain, L Whitlow linker, VL variable light chain, tm transmembrane region, ic intracellular, LTR long terminal repeat. **b** Murine splenic T cells were transduced with the mCXCR5 CAR or GFP, or left UT. CAR surface expression was detected by anti-IgG and anti-CD8 co-staining. Percentages of CD8$^+$CAR$^+$ and CD8$^-$CAR$^+$-T cells are indicated. **c** UT, mCXCR5 CAR- or GFP-transduced T cells ($5 \times 10^4$) were cocultured for 24 h in a 1:1 ratio with murine $E\mu$-$Tcl1$ (CXCR5$^+$), CXCR5-deficient $E\mu$-$Tcl1$, splenic B220$^+$ B cells, CXCR5-deficient BW5147 T cell lymphoma cells, and CXCR5-transduced BW5147 cells. IFNγ and IL2 in the supernatant were quantified by ELISA. Max, CAR-T cells and PMA/Ionomycin; Min, CAR-T cells only. Data of $n = 2$ independently transduced effector cells are shown. **d**, **e** C57BL/6 mice were sublethally (4 Gy) irradiated, followed by i.v. administration of mCXCR5 CAR- or GFP-transduced T cells 4–6 h later in two consecutive experiments: (i) $1 \times 10^6$ transduced T cells were transferred ($n = 3$ mice per group), treated mice and $n = 2$ naive mice were sacrificed at day 7; (ii) $2 \times 10^6$ transduced T cells were transferred ($n = 6$ mice per group). Three mice were sacrificed at day 14 and 20, respectively, naive animals served as controls ($n = 4$). Splenocytes were analyzed for (**d**) percentage of splenic CXCR5$^+$B220$^+$ B cells and of (**e**) CXCR5$^+$CD4$^+$ helper T cells among CD4 T cells. **d**, **e** Data of $n = 2$ experiments and $n = 3$–4 animals per treatment group and time point. $P$ values are calculated for CXCR5 CAR-T cells compared to GFP-transduced T cells by an unpaired two-tailed Student's $t$ test. Further related data to these experiments are shown in Supplementary Fig. 13b and c. **f**, **g** Irradiated (4 Gy) B6 mice were treated with $2 \times 10^6$ mCXCR5 or mSP6 (control) CAR-T cells from congenic mice (CD45.1+, Ly5.1; SJL). 18 days later, blood samples were analyzed for (**g**) B cells (CD45.2$^+$/CD3$^-$/CD19$^+$CXCR5$^+$) and (**h**) CAR-T cells (CD45.2$^-$/CD3$^+$/IgG$^+$) and the mice were immunized with $5 \times 10^6$ SV40 Tag$^+$ 16.113 tumor cells (i.p.). **i** On day 27, antigen-specific T cells in the spleen were analyzed by flow cytometry using a PE-conjugated Kb-Tag IV dextramer (B220$^-$/CD8$^+$Db-Tag IV$^+$). Bars represent means ± SEM. Dots represent individual mice. Treatment groups were $n = 5$, control groups were $n = 3$ (Ctr B6) or $n = 4$ (naive); $P$ values are calculated for mCXCR5 CAR-T cells compared to mSP6 CAR-T cells by a two-tailed Mann–Whitney $U$ test. Source data are provided as a Source Data file.

CXCR5 CAR-T cell group (Fig. 9h). In summary, mCXCR5 CAR-T cells can access the areas of malignant B cell accumulation in B cell follicles in a syngeneic mouse tumor model, resulting in an efficient elimination of leukemia cells.

## Discussion

This study identifies the chemokine receptor CXCR5 as an alternative and safe target for immunotherapy of B-NHL. CXCR5 is not only expressed on mature B cells and their malignant counterparts, but also expressed in the lymphoma niche by Tfh cells that are reported to sustain lymphoma progression foremost in FL and B-CLL[36-38]. CXCR5 is particularly suitable as target in CAR-T cell therapy of lymphoma because of a threefold therapeutic mechanism, namely (i) prevention of tumor cell homing, (ii) direct lymphoma cell elimination, and (iii) abrogation of a tumor-promoting stroma function. As a prerequisite our CXCR5 CAR exhibits high binding avidity; <200 target molecules per cell were sufficient to trigger T cell effector functions. Notably, reactivity of CXCR5 CAR-T cells was recorded against B-NHL cell lines and a variety of primary B-NHL cells. We and others showed that CXCR5 expression is particularly high on lymphoma entities that depend on cognate interactions with the stromal microenvironment in lymphoid tissues, i.e. FL, CLL, and MCL[2,24,25,39-43]. In accordance, CXCR5 CAR-T cells showed a higher reactivity toward these lymphoma entities in vitro in comparison to CD19 CAR-T cells.

Because CXCR5 is also expressed on Tfh cells, CXCR5 CAR-T cells offer an unprecedented opportunity to target the TME as well, a feature that was so far attributed to TCR-mediated effects on cross-presented tumor peptides in stromal cells. This expands current concepts on the functionality of CAR-T cells, which are believed to target exclusively tumor cells. Tumor-promoting functions of Tfh cells were demonstrated in in vitro and in situ analysis of primary material from B-CLL and FL patients[37,38]. Tfh-derived cytokines, foremost IL-4, and IL-21 together with CD40L signals induce antigen-independent proliferation of CLL cells and may contribute to the generation of cells resistant to conventional chemotherapy[37,44]. An important role of the IL-4-dependent Tfh-B cell axis has also been identified in the FL tumor cell niche;[45] a functional Tfh subset within this niche was characterized by CD10 positivity and an IL-4$^{hi}$IFNγ$^{lo}$TNF$^{hi}$IL-21$^{hi}$ cytokine profile[46]. FL-derived Tfh displayed gene overexpression for $TNF$, $LTA$, $IL$-$4$, and $CD40L$, the latter two signals can rescue lymphoma cells from spontaneous and rituximab-induced

apoptosis in vitro[38]. CXCR5$^+$CD4$^+$ T cells in the blood share functional properties with nodal Tfh cells and represent their circulating memory compartment[47,48]. These circulating Tfh (cTfh) cells were also described in autoimmune diseases such as systemic lupus erythematosus (SLE) and juvenile dermatomyositis. Increased numbers of cTfh cells were also found in CLL and DLBCL patients and evidences indicate that they support malignant B cell survival[37,49].

We propose that CXCR5 CAR-T cells will abrogate cTfh and follicle-located Tfh cells together with malignant B cells and by that, simultaneously target the TME and lymphoma cells. Our data support this notion by showing (i) antigen-specific activation of CXCR5 CAR-T cells in co-cultures with human CXCR5$^+$CD4$^+$ peripheral blood T cells, (ii) complete depletion of cTfh cells in FL and CLL samples from peripheral blood by co-cultures with CXCR5 but not with CD19 CAR-T cells, and (iii) efficient depletion of follicular CXCR5$^+$CD4$^+$ T cells in the spleen of mice treated with anti-murine CXCR5 CAR-T cells.

There is an emerging link between Tfh cells and subtypes of T-NHLs, foremost the angio-immunoblastic T cell lymphoma and peripheral T cell lymphoma that share surface markers of Tfh cells and recurrent genetic abnormalities[50]. We envision that CXCR5 CAR-T cell therapy may also be suitable for the treatment of CXCR5$^+$ T-cell lymphoma entities because of the developmental relationship between Tfh cells and these T cell malignancies, both sharing CXCR5 as a potential target.

The anti-CXCR5 CAR showed no reactivity against immature B-NHLs and precursor B cell neoplasia. This suggests that also non-neoplastic B cell precursors are spared, which contrasts to the activity of CD19 or CD20 CAR-T cells. We expect that the dominating therapeutic side effect of CXCR5 CAR-T cell administration is the elimination of mature B cells and of Tfh cells.

A recent study indicated the presence of CD8$^+$CXCR5$^+$ T cells in FL and proposes that this functional subset mediates anti-tumor activity[51]. In view of their very low numbers in FL as well as in carcinoma, a postulated protective role lacks evidence of clinical efficacy. We do not rule out an anti-tumor reactivity of this CD8$^+$ T cell population, but this reactivity may play a role in other stages of tumor development, because it seems not sufficient to control a large established or progredient lymphoma. We envisage that eliminating a large tumor mass by our CXCR5 CAR-T cells is of higher therapeutic benefit than the loss of a minor tumor-infiltrating CD8$^+$CXCR5$^+$ T cell population.

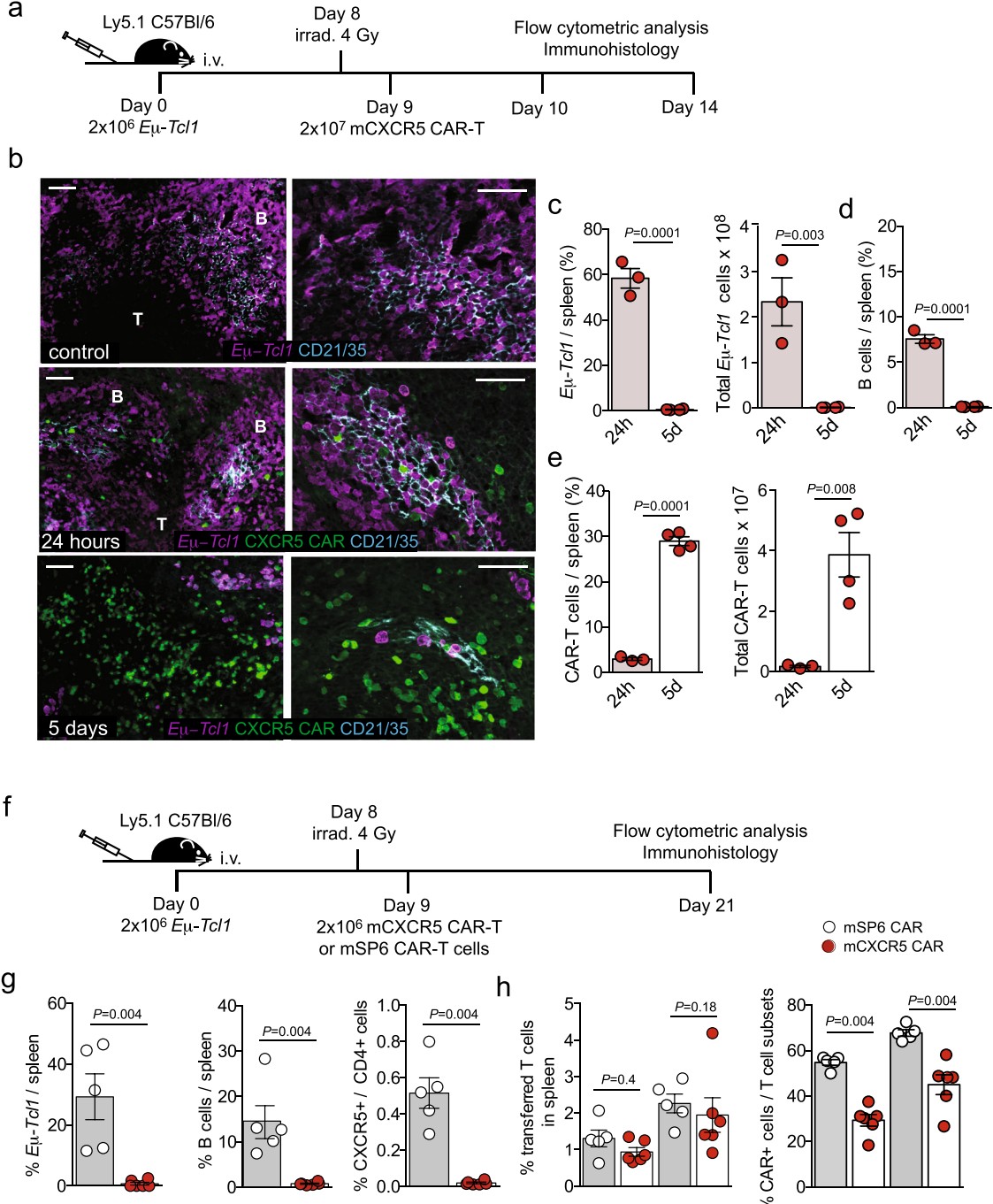

**Fig. 9 Anti-murine CXCR5 CAR-T cells gain access to tumor cells in the B cell follicles and exert potent anti-lymphoma activity. a–e** Congenic mice were injected with $2 \times 10^6$ primary Eμ-Tcl1 lymphoma cells 8 days later, the mice were irradiated (4 Gy) followed by i.v. administration of $2 \times 10^7$ CAR-T cells expressing mTurquoise2 24 h later. In two consecutive experiments mice were sacrificed either 24 h or 5 days (5d) thereafter and splenic CAR-T cells (mTurquoise2+/CD45.1+CD3+), Eμ-Tcl1 lymphoma cells (CD19+ CD5+/CD45.2+) and B cells (CD19+/CD45.1+) were analyzed (**b**) by immunohistology (Scale bars, 50 μm) and (**c**, **d**, **e**) by flow cytometry. Spleen sections were stained with an anti-CD21/CD35 antibody to indicate localization of FDC networks in the B cell follicle. Twenty-four-hour group $n = 3$, 5-day group $n = 4$, control $n = 1$ (no CAR-T cells). Additional histologies are shown in Supplementary Fig. 14. (**f**) Congenic mice (CD45.1+, Ly5.1, SJL) were injected with $2 \times 10^6$ primary Eμ-Tcl1 lymphoma cells and irradiated (4 Gy) 8 days later, followed by i.v. administration of $2 \times 10^6$ mCXCR5 or mSP6 (control) CAR-T cells the next day. **g**, **h** On day 21, splenocytes were analyzed by flow cytometry for Eμ-Tcl1 lymphoma cells (CD45.1−/CD19+CD5+), B cells (CD45.1+/CD3−/CD19+), CXCR5+ helper T cells (CD45.1+/CD3+/CD4+/CXCR5+) and transferred CAR T-cells (CD45.2+/CD4+/IgG+ and CD45.2+/CD8+/IgG+). mSP6 CAR group $n = 5$, mCXCR5 CAR group $n = 6$. $P$ values are calculated for CXCR5 CAR-T cells compared to SP6 CAR-T cells by a two-tailed Mann–Whitney $U$ test. Source data are provided as a Source Data file.

Conflicting data have been reported regarding CXCR5 expression on innate immune cells as well as non-hematopoietic cells of the CNS under inflammatory conditions. A role of the CXCL13/CXCR5 axis was proposed in a mouse model of

neuropathic pain in which neuronally produced CXCL13 activated astrocytes via their increased CXCR5 expression[52]. However, RNA-sequencing data generated from mouse astrocytes after severe spinal cord injury (SCI) showed contradictory

results[53]. We revisited published data via the Astrocyte Reactivity RNA-Seq Browser and compared the expression profiles of CXCR5 and the astrocyte marker GFAP. We found that uninjured as well as SCI astrocytes do not express CXCR5, whereas GFAP is expressed in uninjured astrocytes and strongly upregulated upon SCI. In line with these reports, our extensive on-target/off-tumor toxicity analysis using primary human cell types demonstrated that, overall, CXCR5 CAR-T cells exhibited no cross-reactivity and no on-target/off-tumor effects against a wide range of cell lines or primary cells derived from critical tissues. Even the induction of IFNγ-mediated inflammatory condition did not alter the responsiveness of CXCR5 CAR-T cells against antigen-negative cell types.

Tumor relapse in our xenotransplantation model coincided with low numbers of persistent human CAR-T cells. At the time of relapse, CXCR5 antigen was still expressed on remaining tumor cells, which contrasts to the results from anti-CD19 CAR-T cell-treated lymphoma-bearing mice[54]. Furthermore, tumor relapse might occur due to CXCR5 CAR-T cell exhaustion. However, expression of the exhaustion markers PD-1, LAG-3, and TIM-3 were not upregulated on CXCR5 CAR-T compared to SP6 CAR-T cells when we analyzed the persistence of CXCR5 CAR T cells in vivo at the time of lymphoma cell outgrowth.

We think it is conceivable that tumor relapse may be attributed to the limitations of the xenotransplantation model in NSG mice. More specifically, murine cytokines and growth factors do not sufficiently support long-term expansion of human T cells. In an in vitro stress assay, recursive antigen encounter of the CXCR5 CAR-T cells resulted in extended maintenance of proliferative capabilities and anti-tumor effector function. Hence, our CXCR5 CAR construct confers robust persistence compared to previously described CD19 CAR-T cells with the CD28 costimulatory domain[35].

To overcome the limitations of a xenotransplantation mouse model, we generated an anti-murine CXCR5 CAR construct that endowed murine T cells with specific activity against murine leukemia and primary B cells. The conserved expression pattern of the chemokine receptor between mouse and men allows a prediction of cross-reactivities and on-target/off-tumor effects in a syngeneic mouse model. Upon adoptive transfer of mCXCR5 CAR-T cells, effective antigen-specific ablation of splenic B cells, malignant B cells, and Tfh cells occurred. Notably, off-target toxicity was undetectable after CAR-T cell transfer suggesting that undesired tissue expression of CXCR5 is unlikely. Most importantly, the syngeneic mouse tumor model allows to study mCXCR5 CAR-T cell dissemination and functionality in an immunocompetent lymphoid microenvironment. We succeeded to show that mCXCR5 CAR-T cells gain access to malignant B cells in splenic B cell areas of pre-conditioned tumor-burdened mice and confer efficient anti-tumor reactivity herein.

Beyond the field of cancer therapies, CXCR5 targeting could also be applied for the treatment of autoimmune diseases; we expect selective depletion of mature B cells and of Tfh cells that can promote autoimmunity without impact on precursor B cells. Only recently, CD19 CAR-T cells have been used to treat auto-antibody-mediated diseases such as SLE and pemphigus vulgaris (PV)[55,56]. B cell depletion with rituximab (anti-CD20) showed therapeutic promise in rheumatoid arthritis (RA)[57] and multiple sclerosis[58], but failed to efficiently deplete B cells in SLE and PV[55,56]. Hence, more fine-tuned and longer-lasting approaches targeting autoreactive B cells and auto-antibody responses are warranted.

Taken together, anti-CXCR5 CAR-T cells do not only provide a superior treatment strategy for B-NHL by co-targeting Tfh cells of the TME but also have the capacity to eliminate autoreactive B cells for the treatment of autoimmune diseases.

## Methods

**Construction of anti-CXCR5 chimeric antigen receptors.** The rat CXCR5 hybridoma RF8B2[29] was sequenced and rat germline residues in the framework regions of the variable heavy (VH) and light chain (VL) were replaced for their human counterparts. A single-chain fragment variable (scFv) sequence was designed by adding a IgK leader and by linking the VH and VL sequences through a Whitlow sequence[31]. The CAR backbone consists of a human IgG1 (266-503 aa) spacer with mutated Fc-γ binding regions[59], a CD28 transmembrane domain, an intracellular CD28 costimulatory domain and a CD3ζ activation domain. CD28 and CD3ζ were derived from a CD19 CAR construct[60] or NCBI RefSeq database. The codon-optimized CXCR5 CAR was synthesized (GeneArt; Thermo Fischer Scientific) and cloned into the retroviral vector MP71[61]. Deletion of the intracellular CD28 costimulatory domain from this construct resulted in a first-generation anti-CXCR5 CAR. Primer sequences are presented in Supplementary Table 1. As controls we used a SP6 CAR recognizing the hapten 2,4,6-trinitrophenyl and a CD19 CAR, which is derived from the murine anti-CD19 mAb FMC63[31].

The anti-mouse CXCR5 CAR had a similar design. The rat CXCR5 hybridoma 2G8[20] was sequenced and a codon-optimized scFv sequence was introduced into a murine CAR backbone consisting of an IgG1 spacer (228 aa), a CD4 transmembrane domain and an intracellular CD28 costimulatory domain followed by a CD3ζ activation domain[62]. For some in vivo experiments, the anti-mouse CXCR5 CAR was expressed in tandem with a fluorescent reporter (mTurquoise2) using a P2A element.

**Mapping of the anti-CXCR5 mAb epitope.** The epitope recognized by the anti-human CXCR5 mAb RF8B2 was mapped using Jurkat cell lines expressing CXCR5 deletion variants and GFP. The full-length CXCR5 (wt) and an isoform with a deleted N-terminus (Δ2−48) were amplified from cDNA clones (OHu17888, OHu18933; GenScript/Biozol) and a P2A-GFP sequence was attached by PCR. Δ2−8 and Δ2−18 variants were generated using forward primers attaching a start codon to the N-terminus. To generate Δ22−45, Δ30−45, and Δ38−45 variants, N- and C-terminal fragments were amplified from CXCR5-P2A-GFP and fused together. Primer sequences are presented in Supplementary Table 1. All variants were cloned into MP71 vectors, and transduced into Jurkat cells. The binding of the mAb to the GFP-positive cell population was analyzed by flow cytometry.

**Avidity analysis of CXCR5 CAR-T cells.** Jurkat cells were stably gene-modified using a transposon vector system (SB100X) to express the human *CXCR5* gene[63]. Single-cell clones were generated by limiting dilution and surface CXCR5 protein was quantified by flow cytometry using Quantibrite PE calibration beads (BD Bioscience). CAR-T cells were co-cultured with clones expressing graded amounts of CXCR5 and IFNγ release was quantified by ELISA (BD Bioscience).

**Retroviral vector production.** Human CXCR5, CD19, and SP6 CAR viral supernatants (SN) were produced by transient transfection[61], or stable producer cell lines were generated by transduction of 293VecGalV cells with MP71-CAR retrovirus. For transduction of mouse T cells, ecotropic retrovirus particles were produced either by transient transfection of the Platinum-E packaging cell line[64] with MP71 plasmids, or by using stable producer cell lines based on transduced Platinum-E packaging cells.

**Primary B-NHL patient samples and peripheral blood cells from healthy donors.** Twenty-five primary B-NHL samples were derived either from PBMC (FL #252, #457, #5632; MCL #114, #578, #624, #988; MZL #421, #441, #769, #728; CLL #12, #31, #33, #47, #75, #819) or from BM (FL #85, #1100, #1054; MCL #545, #777; CLL #890, #821). The study was conducted according to the Declaration of Helsinki and in accordance with local ethical guidelines; written informed consent of all patients was obtained. Usage of primary tumor samples and PBMCs in this study was approved by Ethikausschuss 1 am Campus Charité - Mitte (EA1/003/17; EA1/222/13; EA1/207/14) and Ethikausschuss am Campus Virchow-Klinikum (EA2/216/18). B-NHL subtypes were diagnosed by expert Charité-Universitätsmedizin Berlin hematologists and pathologists.

**Mice.** Eμ-Tcl1, and Cxcr5−/−Eμ-Tcl1 transgenic mice on a C57BL/6 background[27] and C57BL/6 and congenic SJL (Ly5.1, CD45.1) mice were bred in-house in a SPF environment (temperature: 22 ± 2 °C; humidity: 55 ± 10; 12-h light/ 12-h dark cycle). NSG (NOD.Cg-Prkdcscid Il2rg tm1 Wjl/SzJ) breeding pairs were obtained from Jackson ImmunoResearch Laboratories and NSG mice for experiments were bred in-house in a SPF environment. All animal studies were performed according to institutional and Berlin State guidelines of the "Landesamt für Gesundheit und Soziales" (LAGeSo) Berlin (G0373/13; G0050/16; G0104/16; G0331/19).

**Lymphoma xenotransplantation models in NSG mice.** The human MCL cell line JeKo-1 expressing firefly luciferase and eGFP was generated by lentiviral transduction[31]. Tumor cells were injected intravenously (i.v.) and their growth was monitored weekly by detecting bioluminescence signals using the IVIS Spectrum imaging (Perkin Elmer) after luciferin i.p. application. CAR-T cells were injected i.v. as indicated in the Figure legends. Living Image Version 4.5 software (Perkin

Elmer) was applied to analyze the bioluminescence average radiance for each mouse as photons/s per cm$^2$ per steradian and binning and exposure were adjusted to achieve maximum sensitivity without leading to image saturation. At the end of the experiment, the presence of human CAR-T and tumor cells was determined in BM and spleen by flow cytometry.

**Syngeneic CAR-T cell transfer model in mice.** C57BL/6 and congenic SJL mice (10–14-weeks old) were used as T cell donors and recipients. Mice were irradiated (4–4.5 Gy) 12–24 hours before adoptive transfer of $1 \times 10^6$, $2 \times 10^6$ or $2 \times 10^7$ CAR-T cells per animal, as indicated. In some experiments, mice were transplanted i.v. with $2 \times 10^6$ tumor cells from the spleen of Eµ-Tcl1 mice 7–8 days prior to irradiation and CAR-T cell transfer. Immunization of mice was performed 18 days after CAR-T cell transfer by i.p. injection of $5 \times 10^6$ live SV40 large T antigen-expressing 16.113 tumor cells[65]. Timepoints at which blood samples were analyzed or animals were sacrificed are indicated in the figure legends. Serum samples were analyzed using a AU480 chemistry analyzer (Beckman Coulter).

**Assessment of VCN.** VCN per cell was assessed according to a protocol kindly provided by Dr. Michael Rothe, Hannover Medical School, Germany. In brief, we determined the absolute number of the woodchuck hepatitis virus post-transcriptional regulatory element (wPRE) encoded by the provirus and of the endogenous polypyrimidine tract binding protein 2 (PTBP2) by qPCR. Genomic DNA (gDNA, 100 ng) from CXCR5 CAR-, SP6 CAR- or untransduced T cells was isolated (Invisorb Spin Tissue Mini Kit, Stratec) and amplified with primers and probes specific for the wPRE element and PTBP2 (Eurofins Genomics) using the Applied Biosystems StepOnePlus Real-Time PCR Systems (conditions: 50 °C for 2 min, 95 °C for 20 s, followed by 40 cycles of 95 °C for 5 s, 56 °C for 20 s, 65 °C for 30 s (primer conc.: 330 nM; probe conc.: 150 nM; ABI TaqMan Fast Advanced Master Mix from Thermo Fisher Scientific). Primer and probe sequences are presented in Supplementary Table 1. Absolute copy numbers were calculated using parallel amplification of $10^3 - 10^6$ copies of a linearized standard plasmid (pQPCR-Stdx4) with StepOne software (v2.3) (Thermo Fisher Scientific). The VCN per transduced cell was determined by normalizing to PTBP2 and the transduction rate.

**Human T cell transduction.** The study was conducted according to the Declaration of Helsinki and in accordance with local ethical guidelines; written informed consent of all patients was obtained. An approval for recruiting voluntary, healthy blood donors has been obtained from Ethikausschuss 1 am Campus Charité - Mitte (EA1/003/17; EA1/222/13) and Ethikausschuss am Campus Virchow-Klinikum (EA2/216/18). Human peripheral blood mononuclear cells (PBMCs) were purified using Biocoll solution (Biochrom, Berlin, Germany). PBMCs were cultured in TCM: RPMI, 10% FCS (PAN-Biotech), 1 mM NaPyr, 0.1 mM MEM NAA, 100 U/ml penicillin, 100 µg/ml streptomycin (all Thermo Fisher Scientific). Untransduced control cells were prepared in parallel and subjected to the same protocol except that TCM instead of viral supernatant (SN) was used for transduction. Depending on whether the human T cells were used in in vivo or in vitro experiments, they were cultured either with IL7/IL15 (10 ng/ml each), or IL2 (10 ng/ml), respectively (all Miltenyi Biotec). PBMCs were activated for two days on anti-CD3 (5 µg/ml) and anti-CD28 (1 µg/ml) (both BioLegend) mAb-coated 24-well plates. For transduction, RetroNectin (RN, TaKaRa)-coated plates were prepared as follows: 24-well non-tissue culture plates were incubated with 500 µl/well RN-solution (12.5 µg/ml) overnight at 4 °C. RN solution was removed, plates were blocked (2% BSA) and washed (PBS). On the day of the first transduction, 250 µl/well rapidly thawed viral SN was added and the plates were centrifuged (3000 × g, 2 h, 4 °C). Afterwards, the viral SN was removed, the activated T cells were transferred to the virus-coated plates, 1 ml/well TCM mixed with viral SN (4:1) supplemented with either IL7/IL15 or IL2 was added and the plates were centrifuged (800 × g, 30 min, 32 °C). Next day, a second transduction round was performed. One ml medium per well was removed, 1 ml viral SN (4:1) supplemented with cytokines was added and the plates were centrifuged (800 × g, 30 min, 32 °C). For IL2 cultures, TCM with IL2 (10 ng/ml) was added every day, each time increasing the volume 1.5–2.0 fold. On day 10, the T cells were cultured at a density of $2 \times 10^6$/ml in fresh TCM with low IL2 (1 ng/ml). Three days later, T cells were frozen and stored in liquid nitrogen until their use in in vitro assays. For IL7/IL15 cultures, every other day 1/3 of the medium was removed and the remaining volume was doubled with fresh TCM containing IL7/IL15 (10 ng/ml each). After 10–12 days, T cells were stored in liquid nitrogen until use. For some experiments, an alternative protocol for the transduction of human T cells was employed. PBMCs were purified using Biocoll solution (Biochrom) or T cells were isolated using the Miltenyi CliniMACS Prodigy via magnetic sorting of T cells with anti-CD4 and anti-CD8 beads (Miltenyi Biotec). Subsequently, cells were stored in liquid nitrogen until use. PBMCs or purified T cells were cultured in TexMACS medium supplemented with IL7/IL15 (10 ng/ml each) (TexMACScyt) (all Miltenyi Biotec) and with 1% human AB-serum for the first five days of culture (ZKT). One million cells were activated for two days using 10 µL TransAct (Miltenyi Biotec) in 1 mL TexMACScyt medium. For transduction on day 2, 500 µL medium was removed and 500 µL transduction mix was added and the cells were centrifuged at 32 °C and 400 × g for 2 h. The transduction mix consisted of TexMACScyt + 1%AB, Vectofusin-1, and 150 µL viral supernatant.

After the addition of the transduction mix another 200 µL TexMACScyt + 1%AB was added. Twenty-four hours later, 500 µL medium was removed and fresh TexMACScyt + 1%AB medium was added. From day 5 cells were regularly fed with TexMACScyt + 1%AB medium and harvested at day 13. Absolute numbers of T cells were counted using the Miltenyi MACSQuant flow cytometer.

**Mapping of the anti-murine CXCR5 mAb epitopes.** To map the epitope of the anti-mouse CXCR5 mAb 2G8 (BD Biosciences), full-length mouse CXCR5 cDNA was retrieved from Eµ-Tcl1 tumor cells by RT-PCR, fused to P2A-GFP and cloned into the MP71 vector. An N-terminal deletion variant was generated by PCR, cloned into the MP71 vector and transduced together with the full-length gene into the mouse thymoma cell line BW5147. Transduced cells were analyzed by flow cytometry. Primer sequences are presented in Supplementary Table 1.

**Mouse T cell transduction.** Splenocytes were isolated from C57BL/6 mice, adjusted to $2 \times 10^6$ cells/ml in TCM and activated for one day with 1 µg/ml anti-CD3 mAb, 0.1 µg/ml anti-CD28 mAb (both BioLegend). In addition, the TCM was supplemented with either 10 ng/ml each of mouse IL15/IL7, or in some cases with 50 ng/ml IL15 and 10 ng/ml IL7 (Peprotech). For the first transduction, the activated T cells were adjusted to $1 \times 10^6$ per ml in fresh TCM and the following supplements were added: $4 \times 10^5$ beads/ml mouse T-Activator CD3/CD28 (Thermo Fischer Scientific), mouse IL15/IL7, as indicated. Then, $1.3 \times 10^6$ cells/well were transferred onto virus-coated plates. Preparation of virus-coated plates and both rounds of transduction were performed as described above for human T cells. After the second transduction, the T cells were cultured in fresh TCM with cytokines for three days. Then, the medium was again renewed and the cell density was adjusted to $1 \times 10^6$/ml. After six to seven days in culture, residual beads were magnetically removed and the T cells were used directly for in vivo and in vitro experiments.

**Retro- or lentiviral transduction of cell lines.** Retroviral particles encoding CXCR5 were generated by transient transfection of HEK293T cells with plasmids encoding MP71 vector, 10A1 env, and gag/pol[61]. Lentiviral particles encoding GFP and firefly luciferase (luc) were generated by transfection of HEK293T cells with the plasmids pFU-Luc-eGFP, and the packaging plasmids pLP1, pLP2, and pCMV-VSV-g employing CaPO$_4$ precipitation. The transduction was performed in 24-well non-tissue culture plates, that were coated with 25 µg/ml RetroNectin (TaKaRa) for 2 h at 37 °C. Then $5 \times 10^5$ cells in 1 mL medium, 1 ml viral SN and protamine sulfate were added, and the plate was centrifuged at 800 × g for 90 min and at 32 °C. Two days later, transduced cells with high GFP expression were sorted by FACS, further expanded and stored as frozen aliquots.

**Human cell lines and primary cells from healthy tissue.** The human B-NHL cell lines DOHH-2 (ACC 47) and SC-1 (ACC 558; both FL), SU-DHL-4 (ACC 495) and OCI-Ly7 (AC 688; both diffuse large B-cell lymphoma, DLBCL), JEKO-1 (ACC 553; MCL), the NCI-H929 (ACC 163) cell line (multiple myeloma, MM), RAJI (ACC 319; Burkitt lymphoma), the JURKAT (ACC 282) cell line (acute lymphoblastic leukemia, T-ALL), and the hepatocellular carcinoma cell line HEP-G2 (ACC 180) were purchased from DSMZ (Braunschweig, Germany). The colorectal adenocarcinoma cell line SW-620 (#300466) was purchased from CLS Inc. (Eppelheim, Germany), and the human embryonic kidney cell line HEK293 from Quantum Biotechnologies Inc. (Quebec, Canada). The B cell precursor leukemia cell lines NALM-6 and REH were kindly provided by Dr. Stephan Mathas (MDC, Berlin, Germany). The authenticity of REH was confirmed by a multiplex cell line authentication test (Multiplexion). All cell lines were directly expanded after receipt and aliquots were stored in liquid nitrogen. Human umbilical vein endothelial (HUVEC) and human umbilical arterial endothelial cells (HUAEC) were purchased from PromoCell and expanded in endothelial cell medium over 2–3 passages before usage. A panel of primary cells derived from human healthy tissues (HPNC, human perineurial cells; HA, human astrocytes; HCoEpiC, human colonic epithelial cells; HCerEpic, human cervical epithelium cells; HUC, human urothelial cells; and HN, human neurons) were purchased from ScienCell/Provitro (Berlin, Germany) and cultured in poly-L-lysin-coated (Sigma-Aldrich, Munich, Germany) culture dishes with their respective media over 2–3 passages before use. The luciferase-containing IgA$^+$ JeKo-1 cell line was generated by us in our previous BCMA CAR study following the protocol described above.

**Murine cell lines and primary murine lymphocytes.** We used a modified sub-clone of the AKR thymoma BW5147 that does not express endogenous TCR[2]. Murine splenic B lymphocytes (B220+CD3-) were magnetically sorted from C57BL/6 splenocytes stained with anti-B220-PE mAb (RA3-6B2, BioLegend) using the PE Positive Selection Kit from Stemcell technologies (Cologne, Germany). The 16.113 tumor cell line was isolated from LoxP-Tag mice and obtained from Prof. Dr. Gerald Willimsky (Charité - University Medicine Berlin, Germany) and German Cancer Research Center (DKFZ, Heidelberg, Germany).

**Generation of primary murine Eµ-Tcl1 leukemia cells.** Spleen-derived CD5$^+$CD19$^+$ leukemia cell suspensions from 6–10 months old diseased Eµ-Tcl1

and $E\mu$-$Tcl1Cxcr5^{-/-}$ transgenic mice were prepared by tissue homogenization and depletion of red blood cells.

**Syngeneic short-term transfer of CAR-T cells in E$\mu$-Tcl1 tumor-bearing mice**. C57BL/6 Ly5.1 mice (10–14-weeks old) were used as recipients. C57BL/6 mice (10–14-weeks old) were used as T cell donors. Recipient mice were adoptively transferred with $2 \times 10^6$ E$\mu$-Tcl1 tumor cells, that were generated as indicated. After 8 days, mice were irradiated (4 Gy) 24 h before adoptive transfer of $2 \times 10^7$ mTurquoise2-Reporter-CAR-T cells per animal. Animals were sacrificed 24 h and 5 days after adoptive CAR-T cell transfer.

**Immunization**. C57BL/6 mice (10–14-weeks old) were used as recipients. C57BL/6 Ly5.1 mice (10–14-weeks old) were used as T cell donors. Recipient mice were irradiated (4 Gy) 24 h before adoptive transfer of $2 \times 10^6$ anti-murine CXCR5 CAR-T cells per animal. Immunization of mice was performed 18 days after CAR-T cell transfer by i.p. injection of $5 \times 10^6$ live SV40 large T antigen-expressing 16.113 tumor cells. Time points at which blood samples were analyzed or animals were sacrificed are indicated in the figure legends. Eight days after immunization, mice were sacrificed and analyzed by flow cytometry for the existence of antigen-specific T cells. To identify TAg-specific CD8$^+$ T cells, PE-labeled K$^b$/pIV dextramers (Immudex; Virum, Denmark) were used. In context of the K$^b$-restriction element, peptide IV (VVYDFLKL) is the immunodominant epitope among the SV40 large T-antigen derived antigens.

**Flow cytometry analysis**. All reagents and mAbs for flow cytometric analysis were purchased from Biolegend/Biozol unless stated otherwise. FcR block was performed for 15 min at RT with either 5% AB serum (Sigma-Aldrich), human or mouse TruStain FcX in FACS buffer (PBS, 2% FCS, 10 mM EDTA); this reagent was omitted if an anti-IgG Ab was used. Where appropriate, True-Stain Monocyte Blocker was included. Dead cells were excluded by staining with 7-AAD, Zombie Aqua or AquaDye (ThermoFisher Scientific). Specific staining of the mAbs was verified by isotype control staining. Surface expression of human CARs was determined by flow cytometry using a polyclonal FITC or PE-labeled goat anti-human IgG (#2040-09, Southern Biotech/Biozol, FITC 1:400, PE 1:600) or biotin anti-human IgG (IS11-12E4.23.20, #130-119-858, Miltenyi Biotec, 1:10). The cells were washed twice before subsequent costaining.

Costainings were performed using the following mAbs: PB anti-CD3 (HIT3a, #300330, 1:100); BV421 anti-CD4 (RPA-T4, #300532, 1:200) or APC/Cy7 anti-CD4 (OKT4, #317450, 1:100); AF647 anti-LAG-3 (11C3C65, #369304, 1:50); PE anti-PD-1 (EH12.2H7, #329906, 1:50); BV421 anti-TIM-3 (F38-2E2, #345008, 1:50); PB anti-CD8 (HIT8a, #300928, 1:100), APC anti-CD8 (HIT8a, #300912, 1:100) or PE/Cy7 anti-CD8 (HIT8a, 300914, 1:100); FITC anti-CD45RO (UCHL1, #304242, 1:200); PerCP/Cy5.5 anti-CD45RA (HI100, #304122, 1:200); PE/Cy7 anti-CD197 (G043H7, #353226, 1:200); AF647 anti-CD62L (DREG-56, #304818, 1:200); AF647 anti-CXCR5 (RF8B2, #558113, BD Bioscience, 1:100). Tumor cell lines were stained with PE anti-CXCR5 (51505, #FAB190P-100, R&D Systems/Bio-Techne, 1:10), PB anti-CD19 (HIB19, #302224, 1:200) or BV510 anti-CD19 (SJ25C1, #363020, 1:100). Patient biopsies were analyzed either of two mAb cocktails as described in the figure legend: 1) PE anti-CXCR5 (51505, #FAB190P-100, R&D Systems/Bio-Techne, 1:10), BV510 anti-CD19 (SJ25C1, #363020, 1:100) and PB anti-CD3 (HIT3a, #300330, 1:100). 2) PE anti-CXCR5 (51505, #FAB190P-100, R&D Systems/Bio-Techne, 1:10), PB anti-CD5 (UCHT2, #300624, 1:165), FITC anti-CD3 (HIT3a, #300306, 1:100), and APC anti-CD20 (2H7, #302310, 1:100). PBMCs were analyzed with two mAb panels: 1) lymphocytes: FITC anti-CD4 (OKT4, 1:100), PE anti-CXCR5 (51505, #FAB190P-100, R&D Systems/Bio-Techne, 1:10), PE/Cy7 anti-CD56 (5.1H11, #362510, 1:100), APC anti-CD25 (BC96, #302610, 1:100), APC/Fire750 anti-CD8 (SK1, #344746, 1:100), BV421 anti-PD-1 (EH12.2H7, #329920, 1:50) and BV510 anti-CD19 (SJ25C1, #363020, 1:100). 2) Monocytes and DCs: FITC anti-CD1c (L161, #331518, 1:100), PE anti-CXCR5 (51505, #FAB190P-100, R&D Systems/Bio-Techne, 1:10), PE/Cy7 anti-CD11c (Bu15, #337216, 1:100), APC anti-CD14 (HCD14, #325608, 1:100), APC/Fire750 anti-CD303 (201A, #354236, 1:100), BV421 anti-HLA-DR (L243, #307636, 1:100), BV510 anti-CD19 (SJ25C1, #363020, 1:100) and BV510 anti-CD3 (SK7, #344828, 1:100).

CAR expression on transduced mouse splenocytes was detected using polyclonal PE-labeled goat anti-mouse IgG-Ab (#1030-09, Southern Biotech/Biozol, 1:200). The samples for all following stainings were first blocked with anti-CD16/32 (93). In the syngenic mouse model assessing the safety of anti-CXCR5 CAR-T cell therapy, splenocytes were stained with PE-Cy7 anti-PD1 (RMP1-30, #109110, 1:100), APC anti-CD3 (145-2C11, #100312, 1:200), APC/Fire anti-CD4 (RM4-5, #100568, 1:200), BV421 anti-CXCR5 (L138D7, #145512, 1:200) and V500 anti-B220 (RA3-6B2, #561226, BD Bioscience, 1:200). In the therapeutic tumor mouse model, the CAR-T cells were detected using monoclonal PE/Cy7 anti-mouse IgG1 (RMG1-1, #406613, 1:200). Costainings were performed using AF488 anti-CD45.2 (104, #109816, 1:200), APC/Fire anti-CD4 (RM4-5, #100568, 1:200) and PB anti-CD8 (53-6.7, #100725, 1:200). E$\mu$-Tcl1 tumor cells in the spleen were analyzed using FITC anti-CD45.1 (A20, #109816, 1:200), PE anti-CD5 (53-7.3, #100608, 1:200), APC anti-CD3e (145-2C11, #100312, 1:200), BV421 anti-CXCR5 (L138D7, #145512, 1:200) and BV510 anti-CD19 (6D5, #115546, 1:200). Blood

samples from the mouse immunization experiment were stained with PE/Cy7 anti-mouse IgG1 (RMG1-1, #406613, 1:200), APC/Cy7 anti-CD45.2 (104, #109824, 1:200), AF700 anti-CD8 (53-6.7, #557959, BD Bioscience, 1:200), SparkBlue550 anti-CD4 (GK1.5, #100474, 1:400), BV421 anti-CXCR5 (L138D7, #145512, 1:100), BV510 anti-CD19 (6D5, #115546, 1:200) and BUV395 anti-CD3 (145-2C11, #565992, BD Bioscience, 1:200). The antigen-specific T cells were detected using a PE H-2kb/VVYDFLKL (SV40 Tag IV) dextramer (Immudex) and FITC anti-CD45.1 (A20, #109816, 1:200), APC/Cy7 anti-B220 (RA3-6B2, #103224, 1:200) and PB anti-CD8 (KT15, #MCA609PB, Bio-Rad, 1:200).

Data were acquired mostly on a FACS Canto II flow cytometer running FACS Diva version 6.1.3 and were further analyzed with the FlowJo 10 software (all BD Bioscience). Results were exported from FlowJo and further analyzed using Excel 14.7.3 (Microsoft). Samples from the mouse immunization experiment were analyzed using an Aurora Spectral Cytometer running SpectroFlo version 2.2.0 (both Cytek) and the data were analyzed using FCS Express 7.00.0037 Research (De Novo Software). Cell sorting was carried out on a FACS Aria III or FACS Fusion instrument (BD Biosciences).

**Quantification of membrane-bound CXCR5 and CD19 molecules**. To quantify the absolute number of membrane-bound molecules of CXCR5 or CD19 per cell, QuantiBRITE$^{TM}$ PE calibration beads were used according to the manufacturer's instructions (BD Biosciences).

**Cytokine release**. CAR-T cells or untransduced T cells were cultured with or without tumor cell lines or primary B-NHL cells in a 1:1 ratio ($5 \times 10^4$ cells per well each, $1 \times 10^5$ in total) in TCM for 24 h or as specified in the Figure legends. IFN$\gamma$ or IL2 concentration in the cell-free culture supernatant was determined by ELISA (BD Biosciences) according to the manufacturer's instruction. The maximal release was induced by stimulation of effector cells with phorbol myristate acetate (PMA) (5 ng/ml)/ ionomycin (Iono; 1 $\mu$M), and minimum release represents untransduced T cells incubated without target cells. The optical density was determined using a PowerWave microplate reader and KC4 V3.0 software (both Bio-TEK).

**Epitope blocking assay**. Retrovirally transduced CAR-T cells were co-cultured in T cell medium (TCM) for 24 h with JeKo-1 tumor cells at a 1:1 ratio ($5 \times 10^4$ cells per well each, $1 \times 10^5$ total). Peptides synthesized by Biosyntan (Berlin, Germany) were added to the co-culture at different concentrations. The synthesis was performed using simultaneous multiple peptide synthesis and quality was controlled by mass spectrometry. The CXCR5 N-terminal peptide (21 amino acids) is the epitope recognized by the anti-CXCR5 CAR, the control peptide (17 amino acids) is part of the CXCR5 extracellular loop. Supernatant measurement by ELISA, maximum and minimum release was described earlier under Cytokine release assay.

**Cellular cytotoxicity assay**. The kill of radioactive labeled tumor cells by CAR T-Cells was determined in a standard 4-h chromium (Cr) release assay. Labeling of the target cells was performed by incubation of tumor cells with 20 $\mu$Ci $^{51}$Cr sodium chromate (PerkinElmer, Rodgau, Germany) for 90 min at 37 °C. After two washing steps, the target cells ($2 \times 10^3$ per well) were incubated with transduced T cells at various effector to target ratios (E: T) for 4 h at 37 °C. Transduction rates for the CXCR5 and the SP6 CAR were always in the range of 40–60% of all live T cells. Assay supernatants were quantitated in duplicates for $^{51}$Cr release in a Top $\gamma$-Scintillation Count Reader (PerkinElmer). Maximum release values were assessed by hypotonic target cell lysis, and spontaneous release was determined in target cell supernatant without the addition of CAR-T cells. Specific lysis was calculated as follows: % lysis = (experimental lysis – spontaneous lysis) ×100/ (maximum lysis – spontaneous lysis).

**Immunohistology**. For histopathology of CAR-T cell transplanted mice, liver, lung, and colon were collected at the days indicated, fixed in 4% phosphate-buffered formaldehyde, embedded in paraffin, cut into 5 $\mu$m cross-sections, and stained with Hematoxylin and Eosin (H&E). H&E stained tissue sections were analyzed on a Zeiss Axio Imager M2m microscope, equipped with an Apo Tome 2.0 (Carl Zeiss). Images were obtained with a 10x Plan- Neofluar NA 0.3 objective and digital images were processed with Axio Vision 4.8.2 software (Carl Zeiss). Slides from each individual animal were specifically assessed regarding tissue integrity and PMN infiltration.

For frozen sections, half of the spleen was transferred into freshly prepared Paraformaldehyde-Lysine-Periodate-Buffer (10 mg/mL PFA, 2 mg/mL NaIPO$_4$, 0.1M L-Lysine in phosphate buffer (0.05 M Na$_2$HPO$_4$ filled up with 0.05 M NaH$_2$PO$_4$ to a pH of 7.4)) and fixated for 12 h at 4 °C. After fixation, spleens were washed two times in PBS and dehydrated in 30% Sucrose in PBS (~10 h). Spleens were embedded in TissueTek on dry ice and cut into 6 $\mu$m cryostat cross-sections. Sections were blocked with 10% goat serum in PBS and stained using primary labeled antibodies AF594 anti-murine CD21/35 (7E9, #123426 BioLegend, 1:200); AF594 anti-murine gp38 (8.1.1, #127414 BioLegend, 1:200); AF647 anti-human TCL1 (1-21, #330508 Biolegend, 1:100). Stained tissue sections were analyzed on a Zeiss Axio Imager M2m microscope, equipped with an Apo Tome 2.0 (Carl Zeiss).

Images were obtained with a ×20 Plan-Apochromat NA 0.8 and a ×40 objective Plan-Apochromat NA 0.95. Digital images were processed with Axio Vision 4.8.2 software (Carl Zeiss). Where indicated, pseudocolors were assigned to improve discrimination. Slides from exemplary animals were assessed regarding tissue dissemination of mTurquoise2-Reporter-CAR T cells and co-localization with target tumor cells.

**Statistical analysis**. Results are expressed as arithmetic means ± SEM if not otherwise stated. Values of $P < 0.05$ were considered statistically significant, as determined by the two-tailed unpaired Student's $t$ test, unpaired multiple $t$ test, the Mann–Whitney $U$ test, or the Kruskal–Wallis test where appropriate. The statistic features of GraphPad Prism data analysis software suites were used.

**Reporting summary**. Further information on experimental design is available in the Nature Research Reporting Summary linked to this paper.

## Data availability

The data that support the findings of this study are available from the corresponding author upon reasonable request. The source data for the Figs. 1b, d, e, 2c–g, 3a–i, 4c, d, 5c, 6c, d, 7b–d, 8c–e, 8g–i, 9c–e, 9g, h, and Supplementary Figs. S5a, b, S6, S7b, c, S9c, d, S10, S11c–h, S12a, b, S13c are provided as a Source Data file. Reference sequences of human CD28 (NM_006139.4), CD3z (NM_198053.3), CXCR5 (NM_001716.5), and mouse Cxcr5 (NM_007551.3) were downloaded from the NCBI RefSeq database (https://www.ncbi.nlm.nih.gov/refseq/). The data underlying Supplementary Fig. S3 was downloaded from the HS_AFFY_U133PLUS_2-0 database using the free basic edition of the Genevestigator (Nebion) software (https://genevestigator.com) on November 4, 2019. Source data are provided with this paper.

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

## Acknowledgements
We thank Kerstin Krüger, Kerstin Gerlach (Max-Delbrück-Center for Molecular Medicine, MDC, 13125 Berlin, Germany), and Andrea Ellinghaus (Charité-University Medicine Berlin) for excellent technical assistance. We are thankful to Dr. Ulrich Großkinsky and Dr. Felix Oden for technical advice and helpful discussions. We thank the Flow Cytometry and Animal Phenotyping platforms of the MDC for their technical support. This work was funded by grants from the Deutsche José Carreras Leukämie-Stiftung, the MDC PreGoBio program, the Helmholtz "Zukunftsthema" Inflammation and Immunity (U.E.H.), and the Berliner Krebshilfe (all to U.E.H. and A.R.).

## Author contributions
M.B., J.P., J.B., and J.J.J., developed methodology, conducted experiments, validated assays, analyzed data, and wrote sections of the manuscript. M.Z., H.S., A.W., and V.K. conducted experiments and analyzed data. M.C., H.A., and W.U. provided technical resources and technical advice. J.W. diagnosed and provided primary human tissue specimen. U.E.H. and A.R. were responsible for all aspects of the study, including the conception and design of the experimental approach, planning and coordinating experimental procedures, analyzing data, and composing and writing the manuscript. All authors participated in editing and approval of the final text of the manuscript.

## Funding

## Competing interests
U.E.H., A.R. J.B., and W.U. acknowledge filed patent application WO 2019/038368 A1 "Chimeric antigen receptor and CAR-T cells that bind CXCR5" related to the work disclosed in this paper. All authors declare that they have no competing interests.
