## [Peer Review File · Nature Communications]

Reviewers' comments:

Reviewer #1 (Remarks to the Author):

The authors present in vitro and in vivo data on a novel CAR T construct for treatment of nodal B-cell neoplasia. In contrast to the currently applied CAR T constructs, the authors developed an CXCR5 CAR T cells for adoptive therapy. The authors claim that this construct will overcome some of the obstacles of current CD19 CAR T adoptive therapy. This is achieved by the usage of the novel target antigen CXCR5 which is not only expressed on malignant B cells but also on lymphoma-supportive follicular T helper cells. The data demonstrates that the CXCR5 construct is successfully targeting and eliminating lymphoma cells in vivo. However, the superiority of a CXCR5 construct vs a CD19 construct is not convincingly shown albeit comparative data is included. The following aspects have to be considered and addressed by the authors:

1. The claim that the CXCR5 construct is superior to CD19 CAR T constructs is not comprehensively addressed. The authors should comment on the location of the CXCR5 binding site (in relation to the membrane) also in comparison to CD19 constructs, the affinity and avidity to the binding site. Are the target cells expression CXCR5 and CD19 at the same expression level ? is it actually "biology" or rather "technology" of the novel construct.
2. It would have been really interesting to compare a CXCR5 construct with an CXCR5 chemokine receptor plus CD19 CAR construct versus a conventional CD19 CAR T. Do we need the targeting or do we need the homing receptor ? Do we actually need to eliminate the follicular T helper cells ? does the depletion of these cells prior to infusion equalize the different constructs ?
3. Are soluble factors expected to be an "antigen sink" and bind to the construct ?
4. the number of patients shown for expression level of various antigens is low - the variability high - more data is required.
5. I do not understand why repetitive stimulation with target cells does not affect the expression profile or function of the cells ? why is this expected ? what does this differ from other constructs ? what is the "positive" and "negative" control ? was the same experiment done with CD19 CAR T ?
6. How is the avidity of the construct characterized ?
7. Is there evidence that higher expression levels of the target antigen are beneficial for short or long term function of CAR T cells ? is the high expressing target antigen ideal ?
8. The numbering (a-e) and writing in the figures is very confusing and needs to be changed.
9. Figure 3 e and 3f - gating strategy has to be modified or explained; no gating on CD19 cells ? is PD-1 looked at on CD19 cells ? was progression checked for CD19 negative relapse ?
10. were the transduction efficacies similar between the different constructs ? and the ratio of CD4:CD8 cells prior to adoptive transfer ?
11. Methods have to be given in more detail, e.g. T cell transduction ?, Quantification of CXCR5 molecules ?

Reviewer #2 (Remarks to the Author):

Bunse et al provide a report describing the generation and function of CAR T cells that target CXCR5 on tumor cells. The authors suggest that this is a novel and valuable approach because 1) it targets a marker on many lymphomas that differs from CD19, which can be lost, and 2) it also targets Tfh cells, which may play a role in supporting lymphomas. The authors convincingly demonstrate that they can generate T cells in both humans and mice that specifically target CXCR5+ cells, including multiple CXCR5+ lymphoma lines. The CARs become activated to produce IFN γ and kill. These cells do not recognize CXCR5- cells lines, and in vivo, they reduce the numbers of CXCR5+ cells in mice and suppress expansion of CXCR5+ tumor cells. While the tumors expand eventually, the CAR T cells do not show an exhausted surface phenotype.

The authors convincingly demonstrate the generation and function of CAR T cells that target

CXCR5+ cells. My primary question (#1 below) is whether these CXCR5 CAR cells, which will lack CXCR5 themselves, will be able to migrate to follicles and access lymphoma cells that reside within follicles. Further, if CXCR5 CAR T cells can do this, then further evaluation of the consequences of eliminating all CXCR5+ cells in lymphoid follicles, with some microscopic visualization of this process, would be valuable.

MAJOR COMMENTS:

1) Homing of CXCR5-targeting T cells. The authors suggest that CXCR5 is important for lymphomas to get to lymphoid follicles, where they might be supported by local Tfh cells. The CXCR5 CAR T cells will fully lack CXCR5 expression themselves due to fratricide or CXCR5-expressing CXCR5 CAR T.

a) Without the ability to express CXCR5, will these CAR T be able to home to lymphoid follicles? Will they be able to infiltrate follicles to kill malignant cells that reside there? Or will the CAR T cells only access tumor cells in the circulation and at extrafollicular sites?

b) Can the authors demonstrate that the CXCR5 CAR T cells enter into follicles in the LNs, for example using immunofluorescence microscopy?

c) When tumors recur in the in vivo models, are the tumor cells preferentially within lymphoid follicles within spleen or LN (i.e. having migrated to sites that the CXCR5 CAR T cannot access)?

d) If the CXCR5 CAR T cells can get into lymph node follicles, how do the authors think they do so?

2) The value of targeting CXCR5 vs CD19. The authors convincingly demonstrate that they generate CAR T cells that are specific for CXCR5. The relative efficacy of CXCR5- vs CD19-targeting CARs seems likely to depend on the relative expression of these proteins across patients with potentially targeted lymphomas. Can the authors demonstrate that CXCR5 is expressed more consistently than CD19 on tumor cells from cohorts of patients with B-NHL?

3) Specificity and off target effects of CXCR5 targeting.

a) The authors show lack of CXCR5 on multiple cell types in culture. This is fine but not particularly informative as it is well known that endothelial cells, HepG2, etc do not express CXCR5. The evidence for lack of injury to liver and kidney in vivo is helpful and important. CXCR5 CAR T might be expected to have the greatest effects on lymph node and spleen architecture and function. What happens to the LNs in these mice? Are the follicles destroyed? Microscopy evaluation would be valuable to demonstrate a) are the CAR T cells in the follicles? and b) what happens to the follicles? Destruction of the follicles would be expected to have long term consequences on the ability to make immune responses in the future (although perhaps not more than loss of all CXCR5+ B cells and Tfh cells).

b) The authors rightly note that there are CXCR5+ CD8+ T cells that have been described as a critical source of non-exhausted T cells that respond to checkpoint blockade (Im Nature 2016, He Nature 2016). The authors suggest that loss of a minor CXCR5+ CD8+ population may not be consequential, but this seems overly dismissive to me. It is possible that CXCR5+ CD8s might be essential to kill tumor cells that migrate to follicles.

c) If T cells transiently express CXCR5 in vivo after activation in a normal immune response, would the presence of CXCR5 CAR Ts kill off all newly activated T cells, limiting the ability to make any new T cell response in vivo (even non T-B dependent responses)? Can the authors demonstrate that mice with CAR T can make any new T cell response?

4) Exhaustion. The authors suggest that tumor recurrence does not seem due to exhaustion of the CAR T, but this is inadequately assessed. If the authors recover CXCR5 CAR T cells from mice that have developed tumor recurrence, do those CAR T cells retain functionality (IFN γ , killing)? The in vitro restimulation assays in Figure are not convincing without a positive control to show that the method will induce exhaustion of other T cells. Without some positive control, it's not clear what this experiment tells us.

MINOR:

1) Figure 6 does not seem to provide any additional information beyond Figure 7, except a somewhat longer time course. Was a day 19/20 time point collected for the Figure 7 experiments; if so, this should be shown (It seems that Panel 7d is from Day 19/20)? The comparison to CD19 CAR is useful in Figure 7; in this context Figure 6 adds little.

2) While it is plausible that Tfh cells support NHL B cells in follicles, it seems to me that the concept that Tfh cells are necessarily 'tumor-supportive' is overstated. There is evidence as cited that Tfh cells are associated with poor prognosis and good reasons to think they contribute to pathology; however, this should be stated more cautiously. It is also possible that Tfh cells contribute to the anti-tumor response, for example by supporting CXCR5+ CD8+ T cells through production of IL-21. Further there is likely a balance of CXCR5+ Tfh and CXCR5+ Tfr (both of which will be eliminated). Presumably these cells have opposing roles in influencing tumor cells? Thus eliminating both is not necessarily beneficial. The authors may be correct, but labeling Tfh cells as 'tumor-supporting' in the title, abstract, introduction ignores the uncertainty here.

3) The figures switch back and forth between dynamite plots and individual data point plots; not clear why. Ideally all plots would show the individual data points e.g. Figure 5e-g unless it's impractical.

Point-to-point reply

Reviewer #1 (Remarks to the Author):

The authors present in vitro and in vivo data on a novel CAR T construct for treatment of nodal B-cell neoplasia. In contrast to the currently applied CAR T constructs, the authors developed an CXCR5 CAR T cells for adoptive therapy. The authors claim that this construct will overcome some of the obstacles of current CD19 CAR T adoptive therapy. This is achieved by the usage of the novel target antigen CXCR5 which is not only expressed on malignant B cells but also on lymphoma-supportive follicular T helper cells. The data demonstrates that the CXCR5 construct is successfully targeting and eliminating lymphoma cells in vivo.

However, the superiority of a CXCR5 construct vs a CD19 construct is not convincingly shown albeit comparative data is included. The following aspects have to be considered and addressed by the authors:

1. The claim that the CXCR5 construct is superior to CD19 CAR T constructs is not comprehensively addressed. The authors should comment on the location of the CXCR5 binding site (in relation to the membrane) also in comparison to CD19 constructs, the affinity and avidity to the binding site. Are the target cells' expression CXCR5 and CD19 at the same expression level? Is it actually "biology" or rather "technology" of the novel construct.

Response:

Our aim in this manuscript is to present CXCR5 as an alternative target for CAR-T cell therapy.

We designed the CXCR5 CAR using a well-established second generation CAR format: "scFv-IgG Fc-CD28-CD3z". For comparison, we included a CD19 CAR with the same design in all critical experiments. The revised version of the manuscript contains now additional CD19 CAR data. The anti-CD19 scFv, like in many other CD19 CARs, was derived from the mouse FMC63 hybridoma. **Ghorashian et al. (Nature, 2019)** determined that the FMC63 conformational epitope spans loop 1 (AA 97-107) and 2 (AA 155-156). In addition, they applied surface plasmon resonance and determined the KD of a FMC63-derived scFv to be in the subnanomolar range (0.33 nM).

The affinity of the non-humanized rat anti-human CXCR5 mAb RF8B2 is 0.7 nM as determined by ELISA and purified target protein. We generated Jurkat cell lines expressing CXCR5 variants with deletions in the extracellular N-terminal domain and mapped the binding epitope of the RF8B2 mAb to aa 9-30 in the extracellular domain of CXCR5 (**Supplementary Fig. 5a**).

The differences between CXCR5 and CD19 CAR-T cells observed in our manuscript are more likely caused by the different target antigens than by the CAR constructs. The homeostatic chemokine receptor CXCR5 is a 42 kDa member of the 7-transmembrane spanning G protein-coupled receptor family. CXCR5 is expressed by mature B cells, follicular helper T cells, Burkitt's lymphoma cells, and mediates cell migration to the B cell follicles in the secondary lymphoid organs. The ligand of CXCR5 is CXCL13 (BLC).

CD19 is a type-I transmembrane glycoprotein of 95 kDa that belongs to the immunoglobulin superfamily and is widely expressed on B cells throughout most stages of B-cell differentiation, though its expression is down-regulated during their terminal differentiation

to plasma cells. Hence the structural and signaling nature of these two proteins is completely different, also the number of molecules being expressed on the surface of benign and malignant B cells varies up to 10-times between CD19 and CXCR5. In our original manuscript we already showed the number of CXCR5 molecules being expressed on benign B (mean: 8409 molecules) and on malignant B cell lines (mean between 418 molecules (OCI-Ly7), 1667 (JeKo-1), 2846 (SU-DHL4), 2602 (SC-1), and 5146 molecules (DOHH-2) (Fig. 1b, d, e). In our revised manuscript we now compare the expression levels of CD19 (mean 21641 molecules (OCI-1Ly7), 10940 (JeKo-1), 13392 (SU-DHL4), 4285 (SC-1), and 36092 molecules (DOHH-2) (Supplementary Fig. 7c) with the aforementioned numbers for CXCR5. For the B-ALL cell line NALM-6 we even found around 50.000 CD19 molecules per cell. This cell line is very frequently used as a "standard" target cell line for assessing the activity of anti-CD19 CARs (see numerous papers by the Michel Sadelain group and others). Of note, CXCR5 is not expressed on B-ALL cells at all (Fig. 1b). Expression levels in the range of CD19 are completely unphysiological for chemokine receptors in general, underlining again the different nature of these molecules.

Most interestingly, the difference in expression levels between CD19 and CXCR5 has no impact on the strength of the CAR activity towards these cell lines. As an example: the CXCR5 CAR-T cell reactivity toward JeKo-1 is comparably strong as for the CD19 CAR-T cells *in vitro* (Fig. 2f) as well as *in vivo* (Fig. 6 and Supplementary Fig. 9), although JeKo-1 cells express in average 1667 molecules CXCR5 and 10940 molecules of CD19. This holds true for other B-NHL cell lines as well.

Because the anti-CXCR5 CAR and the anti-CD19 CAR are directed against two completely different classes of molecules, comparing the affinities and avidities of the binding sites does not seem to be particularly informative. We argue that the strength of a CAR T cell response also involves the i) number of target antigens expressed, and ii) density of the CAR receptor on T cells itself (Lim and June, Cell 2017; Walker et al., Mol Ther 2017).

To answer the last part of this question by the reviewer: the CXCR5 CAR is clearly a new and innovative CAR with respect to its biology. Notably, it is not only one of the first CARs directed against a chemokine receptor at all, but the first to target simultaneously lymphoma B cells and cells of the TME, the tumor-supporting Tfh cells. In our view, differences between CXCR5 and CD19 CAR-T cells are more likely caused by the different target antigens than by the CAR constructs. We do not conclude that CXCR5 CAR-T cells are in general superior to CD19 CAR-T cells. Both CAR-T cells performed very similar in a number of *in vitro* assays and in the NSG mouse model. However, there was a clear difference between both CAR T-cells in their response towards primary patient-derived tumor samples in *in vitro* assays.

2. It would have been really interesting to compare a CXCR5 construct with a CXCR5 chemokine receptor plus CD19 CAR construct versus a conventional CD19 CAR T. Do we need the targeting or do we need the homing receptor? Do we need actually to eliminate the follicular T helper cells? Does the depletion of these cells prior to infusion equalize the different constructs;

Response:

These are very interesting suggestions. At this point our manuscript is focusing on the detailed biological characterization of a novel anti-CXCR5 CAR, including the extensive validation of CXCR5 expression on mature B cells, T cells subsets, on B-NHL cell lines, and on primary B-NHL samples, derived from mature B cells, profound anti-tumor efficacy *in vitro* and *in vivo*, accompanied by extensive experiments that proved a lack of off-target

activity. Additionally, we generated an anti-murine CXCR5 CAR that facilitated specific benign and malignant B cell and Tfh cell depletion in-vivo without further off-target activity. We are of course aware of the interesting possibility to generate dual specific CARs, however, this clearly goes beyond the scope of this manuscript.

CXCR5 and CD19 are both mature B cell markers, thus CXCR5 serves foremost as a tumor-associated target antigen comparable to CD19. Because CXCR5 has “homing” functionality, we here target additionally a functional molecule relevant for the pathophysiology of lymphoma. The relevance for the lymphoma B cell is the accessibility of a survival niche, that supports better proliferation and eventually, protection from apoptosis (Höpken und Rehm, JMM 2012; Heinig et al., Cancer Discovery 2014). Thus, downregulation of CXCR5 under selective pressure through anti-CXCR5 CAR therapy seems less advantageous for the tumor; in contrast, loss or downregulation of CD19 under anti-CD19 CAR therapy and immunoselection has apparently no adverse consequences for the tumor cell itself and accordingly, is frequently observed in clinics upon disease relapse. A definite answer on this question cannot be reliably given in any preclinical model, this answer must be reserved for later clinical studies in human.

The last question “does the depletion of these cells prior to infusion equalize the different constructs” can unfortunately not be answered due to inherent biological problems of common animal models.

The in vivo testing of our anti-human CXCR5 CAR was done in NSG mice which do not have T and B cells and no proper lymphoid organs, neither do they have Tfh cells. Hence, this question could not be addressed at all with the human CXCR5 CAR. However, in our second model, where we applied our murine CXCR5 CAR in a syngeneic tumor model, we were able to see depletion of CD4+CXCR5+ T cell subsets in-vivo (revised Fig. 8e, Fig. 9g). Selective depletion of Tfh cells prior to infusion is not possible because there is not a single unique marker for the CD4+CXCR5+PD-1+ICOS+ Tfh cell subpopulation, but Tfh cells are defined by a combination of surface markers that have an overlapping expression pattern with other T cell subsets. To our knowledge, genetic models to delete the Tfh subset selectively in mice are not available, yet.

3. Are solvable factors expected to be an "antigen sink" and bind to the construct?

Response:

In case of the chemokine receptor CXCR5 this is unlikely to be the case. There are no data published yet that would suggest that the N-terminus covering the cognate epitope of the anti-CXCR5 CAR can be shedded and that a shedded solvable CXCR5 part could serve as a “sink” or decoy for the anti-CXCR5 CAR, leading to ablation of the effector CAR T cell response.

However, to experimentally exclude this possibility we synthesized a 21-aa peptide which represents the epitope recognized by the anti-CXCR5 CAR, and a control peptide (17 aa) covering part of the CXCR5 first extracellular loop. The peptides were added at increasing concentrations (shown in **Supplementary Fig. 10**) to a co-culture of JeKo-1 tumor cells with CXCR5 or CD19 CAR-T cells. IFN γ release by the activated CAR-T cells was determined after 24 hours and revealed no epitope-specific inhibition of anti-CXCR5 CAR-T cell activity in the presence of a soluble CXCR5 N-terminal peptide used in a wide concentration range. With these data we feel confident that even in the unlikely event of a shedded N-terminus, no inhibition of anti-CXCR5 CAR functionality would occur.

4. The number of patients shown for expression level of various antigens is low - the

variability high - more data is required.

Response:

We comply with the request of the reviewer and analyzed additional primary B-NHL samples (added to **revised Figure 1** and **revised Supplementary Figure 1 and 2**). Overall, we have now characterized 25 primary B-NHL samples: 7 FL, 8 CLL, 5 MCL, and 4 MZL samples. These numbers we consider representative and supportive for our claim that the anti-CXCR5 CAR recognizes primary tumor tissues as well.

With respect to CXCR5 expression levels, we found that FL B cells exhibited robust CXCR5 expression (1602-2604 molecules per cell), except for one out of seven samples where CXCR5 expression was low (#85: 265 molecules). Eight B-CLL patient samples showed uniformly the highest CXCR5 expression with 3063 up to 7159 molecules per cell. Hence, on primary FL and CLL patient samples CXCR5 expression is not variable but uniformly high.

On the other hand, CXCR5 expression on six MCL patient samples varied from low to high levels (329-4633 molecules), and MZL B cells expressed mostly only minor numbers of CXCR5 molecules, (**revised Fig. 1c, d and revised Supplementary Fig. 1a-d and Fig. 2a-d**).

5. I do not understand why repetitive stimulation with target cells does not affect the expression profile or function of the cells? why is this expected? what does this differ from other constructs? what is the "positive" and "negative" control? was the same experiment done with CD19 CAR T?

Response:

The same repetitive stimulation assay was also done with CD19 CAR-T cells and we have included the data for CD19 CARs together with the data for the CXCR5 CAR in the **revised Figure 7**. We performed five repetitive stimulation rounds and determined T cell functionality and exhaustion. CAR-T cells were utilized at day 14 after start of their cultivation period. During recursive activation cycles, antitumor cytolytic activity, IFN γ secretion, and the proliferative capacity of both the CXCR5 as well as the CD19 CAR-T cells were similarly maintained (**Fig. 7a and b**).

To provide an experimental set up in which T cell dysfunction/exhaustion can occur, we now added an additional serial transfer experiment in which we altered the cell culture conditions so that CAR-T cell dysfunction can be observed in both CAR groups. Moreover, we introduced a first generation CXCR5 CAR-T cell construct lacking the CD28 co-stimulatory domain. As expected, although effective killing occurred, the first generation CXCR5 CAR-T cells did not survive beyond the second round of stimulation and by that, serves as a positive control for exhaustion/dysfunction. We also altered the CAR-T cell to tumor cell ratio, which revealed dysfunction of CXCR5 CAR as well as CD19 CAR-T cells from the third round of stimulation on (**revised Supplementary Fig. 12a**). This was accompanied by enhanced PD-1 expression on CD8⁺ CAR-T cells (**revised Supplementary Fig. 12b**). We conclude from these experiments that the functional capacities of CXCR5 CAR-T cells can be maintained over an extended stimulation period in a manner comparable to CD19 CAR-T cells.

6. How is the avidity of the construct characterized?

Response:

To characterize the functional avidity of CXCR5 CAR-T cells, we generated a panel of clonal Jurkat cell lines expressing CXCR5 as a transgene. We selected six clones that

showed CXCR5 surface densities of 194, 1533, 2563, 4520, 15964 and 44410 molecules per cell. These clones were cocultured with CXCR5 CAR-T cells from three different donors (n=3) and the amount of secreted IFN γ after 18 hours was quantified by ELISA. Unmodified Jurkat cells served as negative control. The half maximal cytokine secretion (EC50) was calculated using a nonlinear regression curve (log [agonist] vs. response; three parameters).The results are shown in **Supplementary Figure 5b** in the revised manuscript. The experiment is described in the corresponding legend and in the Methods section.

7. Is there evidence that higher expression levels of the target antigen are beneficial for short or long term function of CAR T cells? Is the high expressing target antigen ideal?

Response:

The avidity of a CAR for target cells integrates several contributing factors, foremost the number of antigen receptors on the surface, the density of the cognate antigen on the target cell, as well as the affinity of receptors for the tumor-associated cell surface displayed antigen, respectively (Lim and June, Cell 2017). Thus, the CAR T cell response is regulated by target antigen and CAR surface density, as sub-threshold expression of either one results in low anti-tumor efficacy (Walker et al., Mol Ther 2017). The question of whether a high expressing target antigen might be ideal cannot fully be addressed in in vitro models; it might be less relevant for the therapeutic success when considering the expression levels and tumor mass of primary tumor tissues. Here, the numbers of target molecules, either CXCR5 or CD19, are important for the response and cannot be modulated for the sake of the therapeutic success.

However, in primary lymphoma tissues the efficacy of the CAR T cell response is not only determined by the antigen density, but also other factors exist that contribute to functional avidity and efficacy, such as T cell transcriptional maturation, cytokine signaling, accessibility to the tumor niche, and co-stimulation. In this view, clinical experiences on CD19 downregulation or even loss can only teach us that there are sub-threshold levels that render a CAR T cell response ineffective. We believe that more clinical observations are necessary to fully address this question, with more careful analysis of treatment failure in different CAR antigen specificities and in different tumor entities. In vitro, increasing the ratio of tumor cells over effector cells mimics somehow the experimental design of the effector T cell response against virus-infected cells, where exhaustion was observed. However, a simple delineation of the virus condition to the tumor condition seems problematic.

8. The numbering (a-e) and writing in the figures is very confusing and needs to be changed.

Response:

We complied with the reviewers request and restructured all Figures accordingly.

9. Figure 3 e and 3f - gating strategy has to be modified or explained; no gating on CD19 cells? is PD-1 looked at on CD19 cells? was progression checked for CD19 negative relapse?

Response:

Different gating strategies are included; tumor cells were not gated on CD19, as in some cases tumor populations were not homogeneously CD19 positive. In the newly added patient samples, tumor cells were gated on CD5 or CD20, depending on whether they were positive for CD5 or CD20. We did not analyze PD-1 expression on CD19 cells, as PD-1 expression is mostly restricted to T cells.

10. Were the transduction efficacies similar between the different constructs? and the ratio of CD4:CD8 cells prior to adoptive transfer?

Response:

Yes, they are comparable and the data for the CD19 CAR are now added in **Supplementary Fig. 7a, b.**

11. Methods have to given in more detail, e.g. T cell transduction? Quantification of CXCR5 molecules?

Response:

We apologize for this misunderstanding; the detailed description of these methods are described in the Supplementary Methods paragraph. We now refer to this more clearly in our main manuscript.

Reviewer #2 (Remarks to the Author):

Bunse et al provide a report describing the generation and function of CAR T cells that target CXCR5 on tumor cells. The authors suggest that this is a novel and valuable approach because 1) it targets a marker on many lymphomas that differs from CD19, which can be lost, and 2) it also targets Tfh cells, which may play a role in supporting lymphomas. The authors convincingly demonstrate that they can generate T cells in both humans and mice that specifically target CXCR5+ cells, including multiple CXCR5+ lymphoma lines. The CARs become activated to produce IFN γ and kill. These cells do not recognize CXCR5- cells lines, and in vivo, they reduce the numbers of CXCR5+ cells in mice and suppress expansion of CXCR5+ tumor cells. While the tumors expand eventually, the CAR T cells do not show an exhausted surface phenotype.

The authors convincingly demonstrate the generation and function of CAR T cells that target CXCR5+ cells. My primary question (#1 below) is whether these CXCR5 CAR cells, which will lack CXCR5 themselves, will be able to migrate to follicles and access lymphoma cells the reside within follicles. Further, if CXCR5 CAR T cells can do this, then further evaluation of the consequences of eliminating all CXCR5+ cells in lymphoid follicles, with some microscopic visualization of this process, would be valuable.

MAJOR COMMENTS:

1) Homing of CXCR5-targeting T cells. The authors suggest that CXCR5 is important for lymphomas to get to lymphoid follicles, where they might be supported by local Tfh cells. The CXCR5 CAR T cells will fully lack CXCR5 expression themselves due to fratricide or CXCR5-expressing CXCR5 CAR T.

a) Without the ability to express CXCR5, will these CAR T be able to home to lymphoid follicles? Will they be able to infiltrate follicles to kill malignant cells that reside there? Or will the CAR T cells only access tumor cells in the circulation and at extrafollicular sites?

Response:

The reviewer raises an important point. Under physiological conditions where secondary lymphoid organs exhibit an undisturbed microarchitecture with a proper B cell follicle and T cell areas, this could potentially be indeed a problem. In our previous publication by **Heinig et al., Cancer Discovery 2014**, we demonstrated that benign B as well as E μ -Tcl1 B leukemia cells need CXCR5 expression to enter B cell follicles where they get in close

contact to follicular dendritic cells (FDCs), the most important follicular stromal cell network.

However, the situation in a pre-treated or disease progredient B-NHL patient is very different. (1) Lymphoid tissue structures are already disturbed by former chemotherapies; (2) tumor patients are pre-conditioned with fludarabine/cyclophosphamide before they receive an adoptive CAR-T cell transplant. This is necessary to expand the niche for CAR-T cells to proliferate and gain uncompeted access to survival factors, e.g. IL-7 or IL-15, and to survive. Consequently, patients are lymphodepleted and their lymphoid organ structure is disturbed at various degrees.

13 and 14

To prove that CXCR5 CAR-T cells which lack CXCR5 expression can enter B cell follicles and efficiently kill follicular leukemic cells, we first set up a short term CAR-T cell transfer experiment. 2×10^6 E μ -Tcl1 tumor cells were transferred into congenic Ly5.1 B6 mice. Eight days later, recipient mice were sublethally irradiated and 2×10^7 CXCR5 CAR-T cells were i.v. administered. The sublethal irradiation mimics pre-conditioning of leukemia patients prior to CAR-T cell treatment, but does not eradicate leukemia cells. One and five days after CAR-T cell transfer, the frequencies and anatomic localization of normal B cells, leukemic cells, and mCXCR5 CAR-T cells in the spleen was determined by immunohistology, and by flow cytometry. Stromal mesenchymal cell types were used as markers to delineate microanatomical compartments. As depicted in the novel **Fig. 9** and **Supplementary Fig. 14**, T and B cell compartments were still maintained 2 days after irradiation and 24 hours after CAR-T cell treatment. The FDC network (CD21/CD35⁺) within the B cell follicles (**Fig. 9b**, **Supplementary Fig. 14a**) and the gp38⁺ fibroblastic reticular cell (FRC) network within the T cell zones (**Supplementary Fig. 14b**) were still present, although stromal networks as well as B cell follicles were reduced in size compared to the controls. Benign B and T lymphocytes were essentially gone. Control animals received leukemia cells, but no irradiation or further CAR-T cell treatment. Most interestingly, 24 hours after mCXCR5 CAR-T cell transfer, CAR-T cells closely intermingled with leukemic B cells within the remaining B cell areas, defined by the presence of FDC networks. Five days after CAR-T cell transfer, efficient leukemic cell reduction, residual apoptotic tumor cells, and a profound expansion of CAR-T cells was observed (**Fig. 9b**, **Supplementary Fig. 10a**). B cell areas were dissolved, FDC networks essentially disappeared, and CAR-T cells predominantly expanded within the partially conserved gp38 networks (**Supplementary Fig. 14b**). Flow cytometry data confirmed these results and showed significant reduction of i) percentages and total numbers of leukemic cells (**Fig. 9c**), ii) percentages of endogenous B cells (**Fig. 9d**), and a iii) significant increase in percentages and total numbers of CAR-T cells in the spleen at day 5 compared to 24 hours after CAR-T cell transfer (**Fig. 9e**).

b) Can the authors demonstrate that the CXCR5 CAR T cells enter into follicles in the LNs, for example using immunofluorescence microscopy?

Response:

Please see our detailed answer **under point 1a**

We can clearly show that mCXCR5 CAR-T cells can access the areas of malignant B cell accumulation in splenic B cell follicles, resulting in an efficient elimination of leukemia cells.

Of note, we can only study this effect in splenic B cell follicles and not in LNs due to the fact that E μ -Tcl1 tumor cells lack CD62L expression which is a prerequisite for benign or

malignant lymphocyte homing into the LN via HEVs. Thus, this particular E μ -Tcl1 tumor model is suitable to study active migration of lymphoid cells in B cell follicles of the spleen, but not in LNs (Heinig et al, 2014). This is also the reason why all analysis presented in this manuscript were done in the spleen and not in LNs. To our knowledge, the E μ -Tcl1 tumor model is widely used in the literature (PubMed: 58 entries on this model) and the leading model to study the pathophysiology of indolent lymphoma, but foremost as a surrogate for CLL or SLL, respectively. However, there is currently no better murine model to study B lymphoma cell migration and expansion in B cell follicles.

c) When tumors recur in the in vivo models, are the tumor cells preferentially within lymphoid follicles within spleen or LN (i.e. having migrated to sites that the CXCR5 CAR T cannot access)?

Response:

When we first decided to generate an anti-murine CXCR5 CAR in addition to our human CXCR5 CAR, the murine construct was meant to be a reagent to overcome the limitations of the xenotransplantation NSG mouse model, the well-known flaw of which is a lack of an organized lymphoid microenvironment including B and T cell areas. This syngeneic immunocompetent mouse model allowed us then to study on-target/off-tumor toxicity and anti-lymphoma activity in immunocompetent mice. The binding moiety of the mCXCR5 CAR is based on the rat anti-mouse CXCR5 mAb (2G8), reactive against the N-terminal domain of mouse CXCR5 (Supplementary Fig. 13a). We designed the anti-mouse CXCR5 CAR (referred to as mCXCR5 CAR) similar to the anti-human CXCR5 CAR, however, all modules are of murine origin (Fig. 8a), except for the rat-derived scFv fragment. This CAR construct was sufficient to analyze on-target/off-tumor toxicity (Fig. 8) as well as localization and interaction of CAR-T cells with lymphoma cells within the follicular TME and subsequent killing over a time span from 5 days to 20 days (Fig. 9). However, beyond that time frame mCXCR5 CAR-T cells are not expanding, most likely because they still exhibit a rat-derived scFv fragment which might be strongly immunogenic to the murine immune system. Therefore, we plan to further improve this model by "murinizing" also the scFv part of the mCXCR5 CAR T cell construct to allow long term survival of the CARs. We believe that we answered already important questions in this syngeneic mouse tumor model that could not be addressed with the hu CXCR5 CARs in NSG mice. Long-term application of the muCXCR5 CAR in this syngeneic leukemia model needs to be addressed with a different optimized CAR product. This is beyond the scope of the present manuscript.

d) If the CXCR5 CAR T cells can get into lymph node follicles, how do the authors think they do so?

Response:

Please see answer under **point 1a and b.**

Homeostatic chemokines and their receptors, CXCR5 and CCR7, are important to gain access to regular formed compartments. In some stages and subsets of lymphomagenesis, other non-homeostatic chemokines may take a leading role. For example, it is known that CAR T cells are equipped with the receptor CXCR3 (own observation, and Newick et al., **Molecular Therapy Oncolytics 2016; Karin, Front Immunol., 2020**), which endows them with migration towards the inflammatory chemokine(s) IP10/CXCL10, CXCL11, and CXCL9 (MIG). CXCR3 expression is also a characteristic feature of some B-cell lymphoma cells (Ohshima et al. **Leuk Lymphoma 2003; Suefuji et al., Int J Cancer 2005; Jones et al., Blood 2000; Kato et al., J Am Acad Dermatol 2009**).

2) The value of targeting CXCR5 vs CD19. The authors convincingly demonstrate that they generate CAR T cells that are specific for CXCR5. The relative efficacy of CXCR5- vs CD19- targeting CARs seems likely to depend on the relative expression of these proteins across patients with potentially targeted lymphomas.

Can the authors demonstrate that CXCR5 is expressed more consistently than CD19 on tumors cells from cohorts of patients with B-NHL?

Response:

As shown in **Supplementary Fig. 1a**, CD19 expression is not always uniformly and highly expressed on primary FL samples, i.e. #252, 457, 563, and 1054 (**Supplementary Fig. 1b**), whereas CXCR5 expression is almost on every patient FL sample uniformly expressed (see **Fig. 1C**).

As presented in **Fig. 3**, we observed that the CXCR5 CAR also mediates superior killing of FL, CLL, and MCL lymphoma cells in comparison to the CD19 CAR. Therefore, we analyzed the frequencies of FL, CLL, and MCL cells and T cells in 48 hour co-cultures by flow cytometry (**Fig. 3f-h; Supplementary Fig. 8a-e**). CXCR5 CAR-T cells killed FL, CLL, and MCL cells more effectively than CD19 CAR-T cells and thus, demonstrating superior anti-lymphoma cell activity. Only the MCL cell line JeKo-1 was depleted by CXCR5 and CD19 CAR-T cells at similar rate (**Fig. 3f-h**). Possibly, differences in the homogeneity and height of CXCR5 and CD19 expression on the primary lymphoma cells may play a role in this observation. However, this may not be the only reason for this result. In situ, what would account additionally to treatment efficacy is our observation that CD4⁺PD1⁺CXCR5⁺ Tfh cells within FL and CLL samples were completely depleted by the CXCR5 CAR-T cells whereas the CD19 CAR-T cells had no effect on Tfh cell numbers (**Fig. 3i; Supplementary Fig. 8d-e**). Overall, our results emphasize that CXCR5 is an attractive alternative target for lymphoma entities that cannot effectively be controlled by CD19 CARs. Secondly, Tfh cells as part of the tumor microenvironment are concomitantly destroyed by CXCR5 CAR-T cells.

3) Specificity and off target effects of CXCR5 targeting.

a) The authors show lack of CXCR5 on multiple cell types in culture. This is fine but not particularly informative as it is well known that endothelial cells, HepG2, etc do not express CXCR5. The evidence for lack of injury to liver and kidney in vivo is helpful and important. CXCR5 CAR T might be expected to have the greatest effects on lymph node and spleen architecture and function. What happens to the LNs in these mice? Are the follicles destroyed? Microscopy evaluation would be valuable to demonstrate a) are the CAR T cells in the follicles? and b) what happens to the follicles? Destruction of the follicles would be expected to have long term consequences on the ability to make immune responses in the future (although perhaps not more than loss of all CXCR5⁺ B cells and Tfh cells).

Response:

As the reviewer points out, depletion of B cells results in impaired B cell follicles and this is what we observed in our microscopic evaluation of splenic B cell follicles 5 days after CAR-T cell transfer (**Fig. 9b-e; Supplementary Fig. 14**). As mentioned earlier under response to question 1b), we investigated this effect in splenic B cell follicles and not in LNs due to the fact that Eμ-Tcl1 tumor cells do not migrate properly into LNs upon adoptive transfer. Of note, B cells were also depleted in LNs upon CXCR5 CAR T cell transfer.

With regard to the consequences of B cell depletion on the ability to make immune

responses, I am referring to our answer to question c).

b) The authors rightly note that there are CXCR5⁺ CD8⁺ T cells that have been described as a critical source of non-exhausted T cells that respond to checkpoint blockade (Im Nature 2016, He Nature 2016). The authors suggest that loss of a minor CXCR5⁺ CD8⁺ population may not be consequential, but this seems overly dismissive to me. It is possible that CXCR5⁺ CD8s might be essential to kill tumor cells that migrate to follicles.

Response:

We partially agree with the reviewers opinion and edited the paragraph in our discussion accordingly. Of note, the cited papers from He et al. and Im et al., Nature 2016 by the reviewer refers to a chronic viral infection model and not to a tumor model. The existence of such a population is undisputed, but we argue that "a postulated protective role lacks evidence of clinical efficacy" for lymphomagenesis. Further, the net therapeutic effect of adoptively transferred CXCR5 CAR T cells is apparently much higher than the loss of this minor tumor-infiltrating CD8⁺ CXCR⁺ T cell population (**Discussion, page 20, middle paragraph**). We do not rule out an anti-tumor reactivity of this CD8⁺ T cell population, but this reactivity may play a role in other stages of tumor development, because it seems not sufficient to control a large established or progredient lymphoma.

c) If T cells transiently express CXCR5 in vivo after activation in a normal immune response, would the presence of CXCR5 CAR Ts kill off all newly activated T cells, limiting the ability to make any new T cell response in vivo (even non T-B dependent responses)?

Can the authors demonstrate that mice with CAR T can make any new T cell response?

Response: We comply with the reviewers request and investigated whether CXCR5 CAR-T cell recipient mice would be severely impaired in mounting a regular T cell immune response. To test this, we transferred 2×10^6 mCXCR5 or mSP6 CAR-T cells into sublethally irradiated mice (**Fig. 8f**). At day 19, continued presence of mCXCR5 CAR-T cells and B cell depletion in the host as a readout for their efficacy were confirmed (**Fig. 8g, h**). Next, animals were immunized with SV40 large T antigen (Tag⁺) expressing Co16.113 tumor cells that normally elicit a strong T cell response in immunocompetent animals. Eight days later, we detected antigen-specific, Db-Tag IV dextramer⁺, CD8⁺ T cells in both the mCXCR5 and the mSP6 CAR-T cell group at similar frequencies (**Fig. 8i**). This result proves that mice treated with anti-CXCR5 CAR-T cells are still able to mount antigen-specific T cell responses.

Also, we like to stress the point that in immunocompromised mice, such as RAG mice, Tag⁺ expressing Co16.113 tumors grow rapidly and eventually, animals succumb from the tumor load (**Willimsky et al., Nature 2005**). Here, our animals treated with CXCR5 CAR T cells did not develop any progressive tumors.

4) Exhaustion. The authors suggest that tumor recurrence does not seem due to exhaustion of the CAR T, but this is inadequately assessed. If the authors recover CXCR5 CAR T cells from mice that have developed tumor recurrence, do those CAR T cells retain functionality (IFN γ , killing)? The in vitro restimulation assays in Figure are not convincing without a positive control to show that the method will induce exhaustion of other T cells. Without some positive control, it's not clear what this experiment tells us.

Response:

The same repetitive stimulation assay was also done with CD19 CAR-T cells and we have included the data for CD19 CARs together with the data for the CXCR5 CAR in the **revised**

Figure 7. We performed five repetitive stimulation rounds and determined T cell functionality and exhaustion. CAR-T cells were utilized at day 14 after start of their cultivation period. During recursive activation cycles, antitumor cytolytic activity, IFN γ secretion, and the proliferative capacity of both the CXCR5 as well as the CD19 CAR-T cells were similarly maintained (Fig. 7a and b).

To also provide an experimental set up in which T cell dysfunction/exhaustion can occur, we now added an additional experiment in which we altered the cell culture conditions so that CAR-T cell dysfunction can be observed in both groups, the CXCR5 CAR-T and the CD19 CAR-T cell treated tumor cell culture. Moreover, we introduced a first generation CXCR5 CAR construct lacking the CD28 co-stimulatory domain. As expected, although effective in killing, the first generation CXCR5 CAR-T cells did not survive beyond the second round of stimulation and by that, serves as positive control for exhaustion. In addition, the altered CAR-T cell to tumor cell ratio caused dysfunction of CXCR5 CAR as well as CD19 CAR-T cells from the third round of stimulation on (revised Supplementary Fig. 12a) This was accompanied by enhanced PD-1 expression on CD8+ CAR-T cells (revised Supplementary Fig. 12b).

The most important conclusion of this experiment is that the functional capacities of CXCR5 CAR-T cells can be maintained over an extended stimulation period in a comparable manner as CD19 CAR-T cells.

We feel reluctant to recover residual CAR T cells from NSG mice that have tumor relapse, because NSG mice do not provide species cross-reactive cytokines such as IL-7 and IL-15 that would maintain human CAR T cells for a prolonged time. Therefore, at the end point of analysis there are too few CAR T cells left that could be analyzed, without prior in vitro expansion. The latter process induces signaling and differentiation processes in T cells that are certainly different from T cells taken directly from animals.

MINOR:

1) Figure 6 does not seem to provide any additional information beyond Figure 7, except a somewhat longer time course. Was a day 19/20 time point collected for the Figure 7 experiments; if so, this should be shown (It seems that Panel 7d is from Day 19/20)? The comparison to CD19 CAR is useful in Figure 7; in this context Figure 6 adds little.

For each *in vivo* xenotransplantation experiment one single human T cell donor has been used; we employed a different donor for former Fig. 5, Fig. 6. In R1, and for former Fig. 7, Fig. 6 in R1. We believe that providing data from more than one independent xenotransplantation experiment with the usage of different donors for the preparation of the CAR-T cells, increases the overall robustness of the *in vivo* experiments and conclusions derived thereof. The slightly different time courses are dependent from the kinetics of the lymphoma development in the control (SP6 CAR) group. The experiment presented in Fig. 6 developed faster than in Fig. 5 and had to be terminated earlier according to our animal protection guidelines, as imposed by the Berlin State review board at the Landesamt für Gesundheit und Soziales, Berlin.

2) While it is plausible that Tfh cells support NHL B cells in follicles, it seems to me that the concept that Tfh cells are necessarily ‘tumor-supportive’ is overstated. There is evidence as cited that Tfh cells are associated with poor prognosis and good reasons to think they contribute to pathology; however, this should be stated more cautiously. It is also possible that Tfh cells contribute to the anti-tumor response, for example by supporting CXCR5+ CD8+ T cells through production of IL-21. Further there is likely a balance of CXCR5+ Tfh

and CXCR5+ Tfr (both of which will be eliminated). Presumably these cells have opposing roles in influencing tumor cells? Thus eliminating both is not necessarily beneficial. They authors may be correct, but labeling Tfh cells as ‘tumor-supporting’ in the title, abstract, introduction ignores the uncertainty here.

Response:

We believe that we discuss this topic deliberately in the discussion section (**Discussion, page 18, 19; Lit.: ref. 35, 36, 42-44**). We mainly refer to published data, but it is not within the scope of the manuscript to address experimentally the role of Tfh cells in any more functional detail. However, we partially comply with the reviewers concern and phrased the topic more carefully in the abstract (page 2) and introduction part (page 4, last paragraph).

3) The figures switch back and forth between dynamite plots and individual data point plots; not clear why. Ideally all plots would show the individual data points e.g. Figure 5e-g unless it's impractical.

Response:

We comply with the reviewers request and show individual data point plots where ever it is practical: hence in addition to **Figure 1** all data from the syngeneic mouse model are presented like that (**revised Fig. 8 and 9; supplementary Fig. 13; revised Fig. 7; supplementary Fig. 13**); the old data of **Fig. 6d**, and **Supplementary Fig. 9d**, and the new data of the CD19 CAR-T cells (**supplementary Fig. 7**).

However, Figures, i.e. IFN γ release in the co-culture systems or the killing of primary B-NHLs (**Figure 2f, complete Fig. 3, Fig. 4; supplementary Fig. 11**) with 15-40 data sets in one graph would get very confusing. Here, we decided to show bar diagrams instead. Of note, the source data of all figures are provided as a separate Source data file.

REVIEWERS' COMMENTS

Reviewer #1 (Remarks to the Author):

Marion Subklewe

Reviewer #2 (Remarks to the Author):

The authors have addressed my comments with additional experiments and revisions; I am satisfied with the revisions. I appreciate that the authors have further explored the infiltration of CXCR5-CAR T cells into follicles. I appreciate the additional experiment demonstrating an adequate CD8 T cell response after treatment with CXCR5-CAR T cells. It remains of interest to determine to what extent a CD4-dependent antibody response can be generated in mice previously treated with CXCR5-CAR T cells. This would presumably be substantially impaired. However, I do not think it essential to address this to publish the manuscript.